# Origin and evolutionary malleability of T cell receptor α diversity

Orlando B. Giorgetti[1✉], Connor P. O'Meara[1], Michael Schorpp[1] & Thomas Boehm[1,2✉]

Lymphocytes of vertebrate adaptive immune systems acquired the capability to assemble, from split genes in the germline, billions of functional antigen receptors[1–3]. These receptors show specificity; unlike the broadly tuned receptors of the innate system, antibodies (Ig) expressed by B cells, for instance, can accurately distinguish between the two enantiomers of organic acids[4], whereas T cell receptors (TCRs) reliably recognize single amino acid replacements in their peptide antigens[5]. In developing lymphocytes, antigen receptor genes are assembled from a comparatively small set of germline-encoded genetic elements in a process referred to as V(D)J recombination[6,7]. Potential self-reactivity of some antigen receptors arising from the quasi-random somatic diversification is suppressed by several robust control mechanisms[8–12]. For decades, scientists have puzzled over the evolutionary origin of somatically diversifying antigen receptors[13–16]. It has remained unclear how, at the inception of this mechanism, immunologically beneficial expanded receptor diversity was traded against the emerging risk of destructive self-recognition. Here we explore the hypothesis that in early vertebrates, sequence microhomologies marking the ends of recombining elements became the crucial targets of selection determining the outcome of non-homologous end joining-based repair of DNA double-strand breaks generated during RAG-mediated recombination. We find that, across the main clades of jawed vertebrates, TCRα repertoire diversity is best explained by species-specific extents of such sequence microhomologies. Thus, selection of germline sequence composition of rearranging elements emerges as a major factor determining the degree of diversity of somatically generated antigen receptors.

The transposon/split gene hypothesis[17] posits that in an ancestor common to all jawed vertebrates, a foundational transposition event split an exon of a cell surface receptor-encoding gene, to create proto-variable (V) and proto-joining (J) segments. The split gene could be reassembled into a functional exon, because the tandem inverted repeats flanking the inserted sequence served as recombination signal sequences (RSS) guiding an ancient RAG-like transposase-turned-recombinase[18,19] to create double-stranded DNA breaks and to excise the inserted sequence; the two free ends would be joined together by the general non-homologous end joining (NHEJ) repair pathway[4]. This hypothetical gene is considered to be the founding member of the entire antigen receptor (AgR) gene family of jawed vertebrates[13,14].

It is conceivable that the diversity arising from somatic recombination of V and J elements and the sequence variability at the VJ junction as a result of error-prone NHEJ repair[4] created a unique selective advantage, as it increased the discriminatory power of the fledgling adaptive immune system through distinct and clonally distributed receptor specificities expressed by individual lymphocytes. However, given the unprecedented emergence of the split VJ gene in ancient vertebrates, possibly as a result of a horizontal gene transfer event[19], it is unlikely that appropriate quality control mechanisms pre-existed that could be harnessed to suppress potentially harmful self-reactivity emanating from somatically assembled receptors encoding unpredictable antigen binding sites[8,20].

Here, we explore the hypothesis that the presence of, and selection for, a particular sequence composition of the coding regions near the RSS elements of V and J genes initially curtailed the sequence diversity of reassembled VJ joints. We propose that the target-site duplication (TSD) that accompanied the transposon insertion[21] and thus flanking both V and J elements, served as a microhomology region minimizing, at least initially, sequence variation at the junction after re-assembly of the V and J parts of the split receptor exon by the NHEJ repair process. In this work, we focus on the T cell receptor α-chain (TCRα) gene, TRA. TRA belongs to the VJ class of antigen receptor genes, reminiscent of the presumptive primordial antigen receptor gene[1,17], encodes one of the chains of the canonical heterodimeric αβTCR (ref. 22) and is a near-universal[23] component of adaptive immune systems of jawed vertebrates. Moreover, the large number of V and J elements that are usually found in TRA loci facilitates the detection of conserved sequence patterns. By contrast, the numbers of V and J segments in TRG (which encodes TCRγ) and immunoglobulin light chain genes are usually much lower and not consistent across species (Supplementary

[1]Department of Developmental Immunology, Max Planck Institute of Immunobiology and Epigenetics, Freiburg, Germany. [2]Faculty of Medicine, University of Freiburg, Freiburg, Germany. ✉e-mail: orlandogiorgetti@gmail.com; boehm@ie-freiburg.mpg.de

Figs. 1–5), making these loci less appealing for in-depth phylogenetic comparisons, especially because several species lack *TRG* and *TRD* (which encodes TCRδ) genes[24] (Supplementary Information). In support of our hypothesis, we find that the presence and extent of corresponding microhomologies at the ends of the coding segments of *V* and *J* elements (that is, the remnants of TSD) are subject to evolutionary changes and inversely correlate with the variable degrees of junctional diversity of the resulting *VJ* assemblies in representatives of all clades of vertebrates. Thus, our results offer an explanation as to how the revolutionary process of somatic diversification could have become successfully incorporated into and maintained in the immune system of jawed vertebrates.

## Predictable *tra* repertoire of zebrafish

We began our analysis by comparing the preselection repertoires of zebrafish and mouse as representatives of ray-finned fish and mammals, respectively, which together represent about 60% of all extant jawed vertebrates[25]. We determined the sequence composition of TCRα gene locus (*tra*) clonotypes of selectable and non-selectable alleles of zebrafish heterozygous for a frameshift mutation in the constant region of the *tra* gene (*trac*). About 95% of all *tra* clonotypes emanating from the wild-type allele are in frame and their sequences show the characteristic size distribution with peaks at every third nucleotide position, very similar to the situation in mice and commensurate with the three nucleotides of a codon (Fig. 1a and Supplementary Table 1). However, even in the non-selectable repertoire of zebrafish, more than three quarters of clonotypes show an in-frame *VJ* joint, whereas the fraction of in-frame rearrangements in mice is close to random (35.6% versus the expected 33.3%) (Fig. 1b). This result indicates the possibility that the germline sequences of *V* and *J* elements strongly influence the outcome of rearrangements in zebrafish. As most assemblies are generated as in-frame rearrangements, we might expect to find that the positions of RSS relative to the reading frames of *V* and *J* genes, respectively, are always the same. Indeed, in 99.2% of all expressed *Va* elements, the RSS begins at position 15 when counting from the first nucleotide of the characteristic cysteine codon at their C-terminal ends (Extended Data Fig. 1). Likewise, in 134 out of 137 *Ja* elements (Extended Data Fig. 2 and Supplementary Table 2), the RSS is situated at the same position (phase 2) relative to the reading frame of the *J* element (Extended Data Fig. 3). Thus, RSS elements downstream of *Va* and upstream of *Ja* elements are positioned in a stereotypic fashion.

The most prevalent configuration of assembled zebrafish *tra* sequences arises from joining *V* and *J* sequences that lack non-templated nucleotides, but show loss of nucleotides (often just one nucleotide) at their ends, ensuring that the reading frames of *V* and *J* elements match (Supplementary Table 1). The size distribution of the resulting complementarity determining region 3 (CDR3) sequences predominantly reflects the lengths of *J* elements (Fig. 1c, left panel and Extended Data Fig. 3), again indicating the strong influence of germline sequence composition on the outcome of recombination. In about 65% of zebrafish *tra* assemblies, the sequences of *VJ* joints show signs of microhomology-guided repair[26–33] (Supplementary Table 1). Up to four nucleotides at the joints cannot be unambiguously assigned to either *Va* or *Ja* sequences (black segments in Fig. 1c, left panel). This contrasts with the sequences of CDR3 regions of TCRα gene locus (*Tra*) assemblies in mice, wherein joints showing additions of non-templated nucleotides to recessed ends (Supplementary Table 1) are the rule, without an appreciable contribution of microhomologies (Fig. 1c, right panel). This result indicates that terminal deoxynucleotidyl transferase (TdT) is not active during the assembly of *tra* genes in larval and adult zebrafish, as it has been shown that TdT interferes with microhomology-dependent recombination and affects the sequence composition of CDR3 regions[31,34–36]. By contrast, the extent of combinatorial diversity in zebrafish and mouse repertoires

is similar. We found that 14,524 out of 16,998 possible *Va*–*Ja* combinations are represented in our zebrafish *tra* repertoires (85.4%), and 2,747 out of 3,000 possible combinations in *Tra* assemblies of mouse (91.6%) (Supplementary Table 1). Thus, the difference between *TRA* assemblies of zebrafish and mouse does not lie with combinatorial diversity, but rather with the drastic differences in the nucleotide compositions of *VJ* joints, most probably due to the presence or absence of TdT activity, which is known to be developmentally regulated in zebrafish[37,38].

In addition to assigning each nucleotide in *VJ* joints to the corresponding germline sequences (Fig. 1c), we also determined to which degree the knowledge about the identity of genes reduces the uncertainty ('entropy') at each nucleotide position across CDR3 regions. The contributions of germline-encoded and non-templated nucleotides to the diversity of CDR3 regions differ between zebrafish and mouse (see Fig. 1d for CDR3 regions of 42 nucleotides in length). Overall, the degree of uncertainty in the nucleotide composition of CDR3 sequences of zebrafish is much smaller than that of mice because, in the former, the germline sequences of *Va* and *Ja* elements contribute to a much greater extent to the final CDR3 sequences (Fig. 1d).

The frequencies with which individual sequences are represented in a repertoire are not uniform as determined by the number of cDNA molecules per sequence; these contributions follow a power law[39–41] (Extended Data Fig. 4). This observation forms the basis for a weighted representation of individual clonotypes and affords the possibility of determining the probability of finding a given molecule in the *tra* repertoires of different numbers of animals. Instead of considering clonotypes, we examine the structure of repertoires in terms of molecules, because they are a proxy of lymphocyte clones and their sizes; otherwise, rare sequences would contribute disproportionally to the composition of the repertoire. In our cohort of six wild-type zebrafish, the probability that a randomly picked *tra* molecule is found in the repertoires of all six fish is 44.1%, whereas the probability of finding a given *tra* molecule in only one of the fish is 9.9% (Fig. 1e and Supplementary Tables 1 and 2); in other words, a large part of the *tra* repertoire is shared among different zebrafish. By contrast, the repertoire of TCRβ (*trb*) assemblies has a large private component (57.5%), whereas only 5.0% of all molecules with identical sequence are found in all six fish (Fig. 1e and Supplementary Tables 1 and 3). We conclude that the high degree of germline sequence-directed outcome of *tra* assemblies in zebrafish is mirrored in the high degree of publicity of sequences in the repertoire (Extended Data Fig. 4). Thus, the generation of *tra* clonotypes in zebrafish is highly predictable.

## Microhomology-directed recombination

Close inspection of zebrafish *tra* assemblies indicates the presence of four main patterns of microhomology-directed recombination (Fig. 2a,b and Extended Data Fig. 4). These small patches of identical sequence are located within short distances of the RSS, compatible with the 5-base-pair length of TSD generated by Transib transposons that are thought to be the ancestors of RAG-like transposons[18,19,21,42,43]. Pattern 1 is based on a di-nucleotide homology (TG), whereas the three other patterns are defined by a single characteristic G or C nucleotide. Together, the four patterns explain the sequences about 70% of all expressed *tra* assemblies (Fig. 2c). As patterns 2 and 3 yield the same length, only three different CDR3 lengths dominate the outcome of each *VJ* recombination (Extended Data Fig. 4). Because there are five main length classes of *J* elements (Extended Data Fig. 3), the superposition of recombination outcomes results in the familiar CDR3 length distribution (Fig. 1a and Extended Data Fig. 4). As the combination of *V* and *J* elements and the effect of microhomology-directed recombination define the outcome of an assembled sequence, we can estimate the probability of generation of a particular clonotype. For a fish with 200,000 T cells, 11,283 *tra* assemblies have a probability of more than

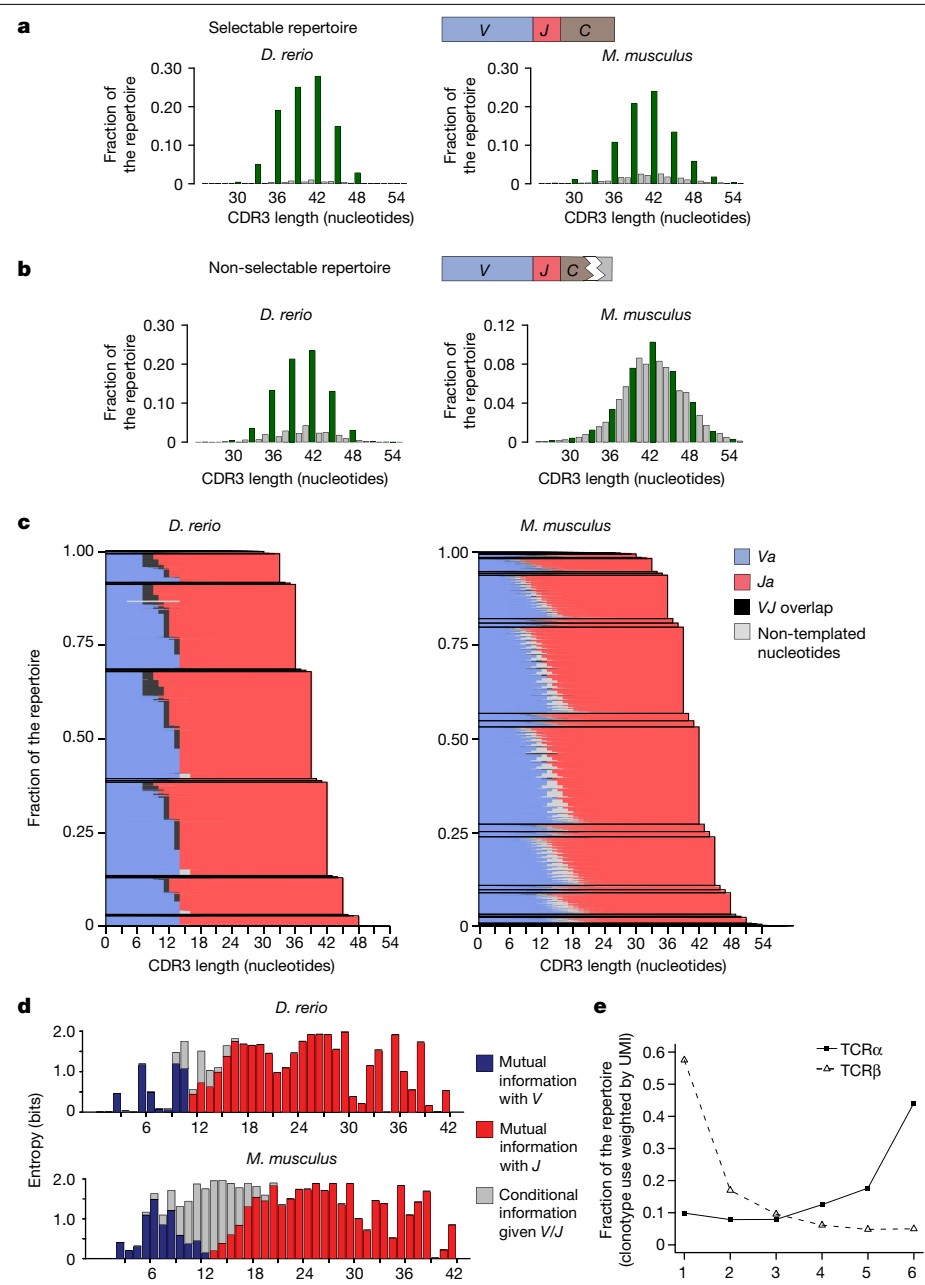

**Fig. 1 | Characteristics of zebrafish and mouse *tra* repertoires. a**, CDR3 nucleotide length distributions for cDNAs of wild-type animals. **b**, CDR3 nucleotide length distributions for cDNAs of animals mutant for *trac* (zebrafish) or *Trac* (mouse) genes. For **a** and **b**, green bars are in-frame rearrangements and grey bars are out-of-frame rearrangements. **c**, Analysis of CDR3 regions of wild-type animals. The origins of nucleotide residues are indicated; variable genes (blue), joining genes (red), ambiguous (either from *V* or *J* regions) (black) and non-templated (grey). The height of lines is proportional to the cDNA count. *Danio rerio*, CDR3 sequences combined from six animals; *Mus musculus*, sequences from three animals. **d**, Diversity at individual nucleotide positions for 42 nucleotide-long CDR3 sequences. Indicated are germline-dependent contributions to mutual information between nucleotide and gene (*V* gene, blue bars; *J* gene, red bars) and germline-independent contributions (conditional entropy when *V* or *J* genes are known, grey bars). **e**, Representation of individual *tra* and *trb* cDNA molecules in 1 to 6 *D. rerio* individuals.

95% for being present in the repertoire (Supplementary Fig. 6); considering the highest number of different *trca* clonotypes that we detected in a single zebrafish individual (57,475 clonotypes in 718,479 molecules), public recombination events correspond to at least 19.6% (11,283 out of 57,475) of the sequence space.

Next, we sought further evidence supporting the conclusion of a largely germline-directed *tra* repertoire. We identified rare instances of zebrafish *Va* and *Ja* elements in the zebrafish genome that show non-canonical positions of the RSS elements. This is exemplified by the

rogue *Va11-6* and *Ja5* elements, which show a deletion or an insertion upstream and downstream of the microhomology regions, respectively (Extended Data Fig. 5). Nonetheless, the stereotypic outcome of microhomology-mediated recombination still prevails, as indicated by the large fraction of out-of-frame assemblies resulting from the use of these elements (Extended Data Fig. 5). Finally, using CRISPR–Cas9-directed mutagenesis, we displaced the canonical position of RSS elements in *Va* genes by a single base pair; as expected for a stereotypical microhomology-mediated recombination, the ratios of in-frame versus

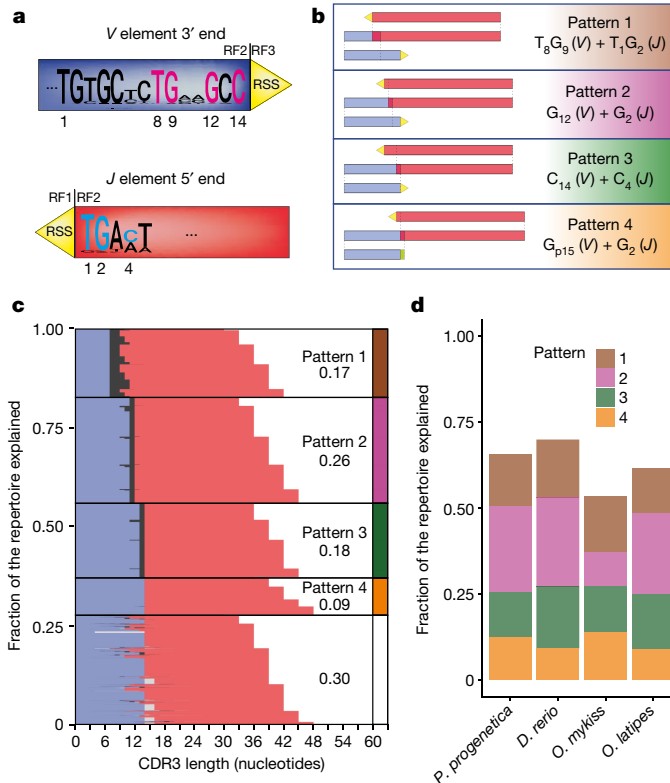

**Fig. 2 | Contributions of microhomologies to the CDR3 region of *tra* assemblies of teleosts. a**, Nucleotide consensus sequences for the 5′ ends of *V* genes (top) and 3′ end of *J* genes (bottom); residues present in the most frequently observed microhomology-directed repair outcomes in *D. rerio* are numbered and coloured. **b**, Schematic of the four most frequently observed patterns of microhomology in *D. rerio*, relative to the conserved positions described in **a**. **c**, Contributions of the four dominant microhomology patterns in **b** to the *tra* repertoire of *D. rerio*. **d**, Contribution of the four dominant microhomology patterns in **b** to the *tra* repertoire in four teleost species.

out-of-frame assemblies were almost inverted (Extended Data Fig. 6). In sum, our experiments indicate that the structure of the assemblies in the zebrafish *tra* repertoire is dominated by microhomology-directed recombination, facilitated by the fixed positions and identities of relevant matching nucleotides.

Microhomology-directed recombination also dominates the *tra* repertoires of three other teleost species (*Paedocypris progenetica*, *Oryzas latipes* and *Oncorhynchus mykiss*), although the relative contributions of the four microhomology patterns identified in zebrafish vary (Fig. 2d). In conclusion, our in-depth analyses of *tra* repertoires of teleost species indicate that the diversity of the *tra* assemblies is, to a surprising degree, predictable and hence establishes a species-specific public repertoire.

## Repertoire predictability

The substantial difference in the magnitude of microhomology-directed recombination of *TRA* genes between teleosts and mouse prompted us to examine this phenomenon in a broader range of species. To this end, we included one representative of cartilaginous fish (brown-banded bamboo shark, *Chiloscyllium punctatum*); two actinopterigian species (Senegal bichir, *Polypterus senegalus*; sterlet sturgeon, *Acipenser ruthenus*) as representatives of basal ray-finned fishes; the West African lungfish (*Protopterus annectens*) as a representative of non-tetrapod lobe-finned fish and, finally, an extra mammal, the African bush elephant *Loxodonta africana*.

We first established consensus sequences for the ends of *Va* and *Ja* sequences in these species (Fig. 3a). The sequence signature supporting microhomology-directed recombination of type 1 ($V\,T_8G_9$ and $J\,T_1G_2$) (Fig. 2a) is well conserved in cartilaginous, basal and teleost fishes; in a substantial fraction of assemblies, the microhomology extends to three and four nucleotides (Fig. 3b). We propose that the 'TG' motif is part of a primordial sequence pattern that directs *tra* assembly in cartilaginous and basal fishes, and teleosts. By contrast, this sequence signature is degraded in lungfish and the two mammals; as a consequence, the contribution of microhomology-directed recombination of type 1 in the *TRA* repertoires of these species is minimal (Fig. 3a,b).

Microhomology-directed assembly generates CDR3 sequences without non-templated nucleotides at the junctions; conversely, diminution of sequence-directed recombination increases their diversity (Fig. 3c). This evolutionary trend can be seen both in the overall structure of *tra* assemblies (Extended Data Fig. 7) and the entropy distributions at the junctions (Extended Data Fig. 8 and Supplementary Table 1).

One consequence of the decreasing effect of microhomology-directed recombination may be relaxed selection pressure on the positions of the RSSs relative to the reading frames of *V* and *J* elements, respectively. For teleosts, which show the highest level of predictability, this notion is reflected in a single peak in the positional distribution of *Va* elements, and the five precisely spaced peaks for the *Ja* elements (Fig. 3d). Shark and basal fishes also show a high degree of positional precision, albeit not as high as the teleost (Fig. 3d). In lungfish, positional acuity is degraded for the *Ja* elements, whereas it is still high for the *Va* elements. By contrast, in mammals, the positions of RSS elements are variable both in *Va* and *Ja* genes. The loss of invariant nucleotide patches at the ends of *Va* and *Ja* elements paves the way for increased diversity of CDR3 regions. In addition to microhomology-directed repair, P-nucleotide-mediated joining and blunt-end joining, both known to be TdT-independent[36], also contribute to the *tra* assembly process. In aggregate, more than 98% of *tra* assemblies of zebrafish can be explained without invoking the activity of TdT (Extended Data Fig. 8). By contrast, *trb* assemblies show substantially higher numbers of non-templated nucleotide insertions at *D–J* and *V–D* junctions (Fig. 3e and Supplementary Table 1). For instance, whereas 93.3% of *tra* sequences of rainbow trout (*O. mykiss*) lack appreciable signs of TdT activity, the substantial nucleotide diversity of *trb* joints is due to TdT activity and approaches that of the mouse (Fig. 3e and Supplementary Table 1). We propose that the difference between *TRA* and *TRB* assemblies with respect to the number of non-templated nucleotides in CDR3 regions is a reflection of species- and stage-specific regulation of TdT activity.

In conclusion, whereas conserved features in *V* and *J* sequences are readily detectable in cartilaginous, basal and teleost fishes, our results indicate that a great change occurred in *TRA* gene structure in lobe-finned fish (Extended Data Fig. 8) that resulted in a near-complete loss of predictability of the *TRA* repertoire in mammals.

## Phylogenetic signatures of *TRA* genes

As discussed above, a key prerequisite for the stereotypic outcome of recombination by means of microhomology-directed repair is the precise positioning of RSS elements next to conserved stretches of nucleotides at the 3′ ends of *Va* and the 5′ ends of *Ja* elements. Moreover, it requires the absence of TdT activity, which is known to reduce the usage of microhomologies[31,34–36]. Erosion of this presumptive primordial pattern is associated with increased diversity of *TRA* assemblies and is first seen for the *Ja* elements (Fig. 3a). This observation suggested the possibility of estimating the extent of predetermined recombination outcomes (and hence the predictability of sequence diversity) by examining the genomic structure of the *Ja* regions also for species for which *TRA* repertoires were not experimentally determined. The same type of analysis is more difficult for *V* regions, because one cannot

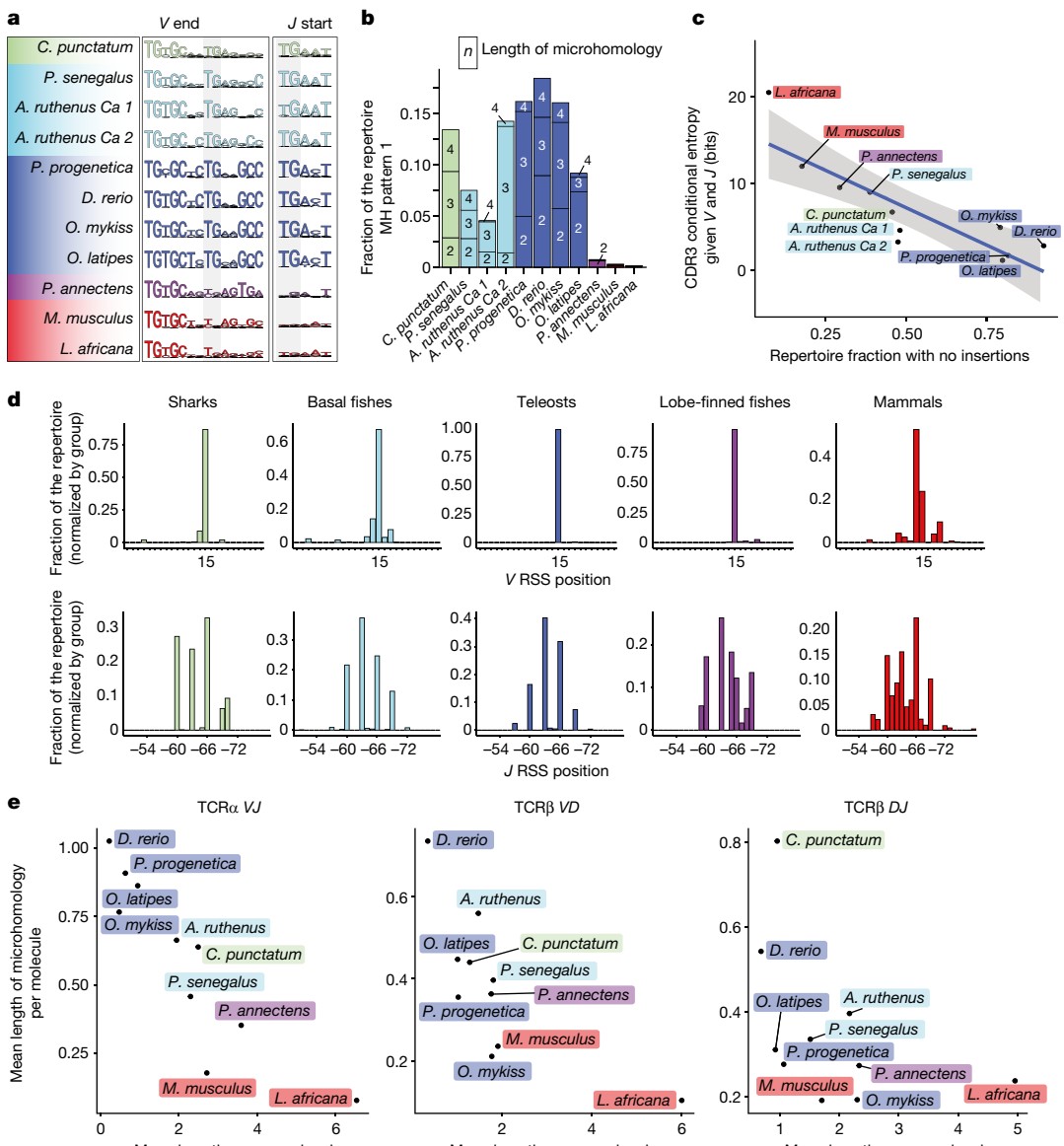

**Fig. 3 | Conserved sequence patterns at the ends of *Vα* and *Jα* genes of jawed vertebrates. a**, Partial nucleotide consensus sequences of *Vα* and *Jα* genes depicted as in Fig. 2a. Nucleotide positions 8 and 9 in *V* sequences, and positions 1 and 2 in *J* sequences are highlighted. The first three nucleotides (TG[T/C]) in the *V* segments correspond to the conserved cysteine residue; in the *J* elements, the distance of the sequences shown to the conserved FGXG tetrad of amino acid residues is variable. **b**, Contributions of microhomologies to CDR3 sequences assembled according to pattern 1 depicted in Fig. 2b; the numbers in boxes refer to the lengths of microhomology segments used. **c**, Correlation between the fraction of *TRA* CDR3 sequences without non-templated nucleotide additions and CDR3 sequence diversity measured as conditional entropy when *V* or *J* sequences are known. **d**, Positions of RSS elements relative to *V* (top) and *J* (bottom) sequences depicted proportionally to their representation in the repertoire. Species counts were normalized and grouped by clades as indicated. For *V* elements, nucleotide position +1 corresponds to the first nucleotide of the characteristic cysteine codon; for *J* elements, nucleotide position −1 corresponds to the thymidine residue of the canonical GT splice donor site at the 3′ end. **e**, Correlation between mean numbers of insertions and mean numbers of microhomology residues for the indicated junctions and species; values are combined for species with several loci.

confidently distinguish *Vα* and *Vδ* elements from genomic sequences alone. Following these considerations, we developed a comprehensive view of *TRA* loci across the main clades of jawed vertebrates by focusing on the presumptive primordial position of RSS relative to the *Jα* reading frame (Fig. 2a and Extended Data Fig. 3). For each of the 302 species in this analysis (Extended Data Figs. 9 and 10), we first calculated the average number of *Jα* elements that conform to the pattern identified in cartilaginous, basal and teleost fishes analysed above (Fig. 3e), and then considered groups of species on the basis of their phylogenetic relationship to determine the group averages. Whereas the levels of conservation of RSS position at the beginning of *Jα* elements are about

the same for cartilaginous, basal and teleost fishes, lobe-finned fishes, reptiles and birds show values that are similar to mammals; amphibians show a broad range of positional averages (Fig. 4a,b and Supplementary Table 4). As expected for the role of microhomology in directing the *Vα*–*Jα* recombination, the summed entropy of the first five nucleotides at the 5′ ends of Jα elements negatively correlates with the deviation from the primordial position of their RSS elements (Extended Data Fig. 9). In conclusion, our broad phylogenetic survey suggests that over a large period of evolutionary time, the sequences of *TRA* assemblies were largely predictable; such predictability then gave way to much higher diversity in lobe-finned fish, setting the stage for the highly

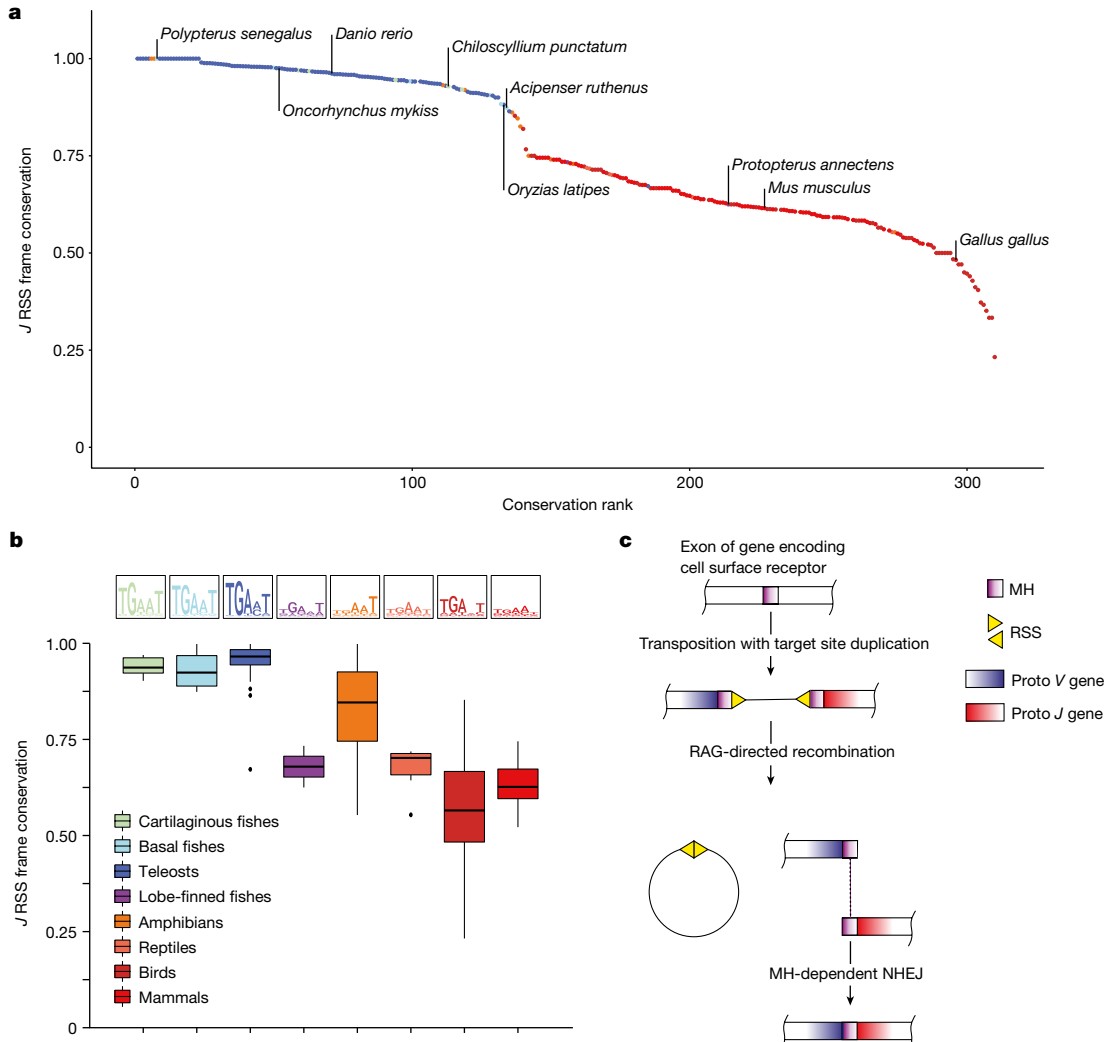

**Fig. 4 | Use of TSDs to guide VJ recombinations. a**, Evolutionary trends for RSS position in *Ja* genes. Species are ordered according to the fraction of *J* genes showing the primordial position of the RSS relative to the reading frame of the *J* gene as depicted in Fig. 2b, with the positions of some species identified. Data originating from duplicated entries (*n* = 8) are not represented. **b**, Distribution of data for the indicated vertebrate clades; the box plots show the median (horizontal line), the first and third quartile (box); the whiskers are drawn for a distance of 1.5 times the interquartile range up to the largest and/or lowest data point from the dataset that falls within this distance. Logo plots depict the first five nucleotides of *J* elements downstream of the RSS. **c**, Scheme depicting the proposed origin of microhomologies at the ends of *V* and *J* genes from TSD after transposon insertion into a primordial antigen receptor gene. MH, microhomology.

diverse *TRA* repertoires characteristic of mammals. As Chondrichthyes and Osteichthyes (including bony fishes and tetrapods) constitute paraphyletic groups[44], we conclude that in the ancestor of jawed vertebrates, microhomologies dominated the outcome of RAG-mediated recombination of antigen receptor genes.

## Discussion

Around half a billion years ago, the facility of somatic diversification of antigen receptors emerged in vertebrates such that each individual acquired the capability to generate a unique (if partially overlapping with other individuals) set of antigen receptors[13,14]. Germline-encoded antigen receptors, which underlie self and non-self discrimination in innate immune systems, are subject to Darwinian selection over evolutionary time. The sudden emergence of the facility of somatic recombination in ancestral vertebrates raises the question as to how the ancestors of vertebrates struck the balance between immunologically desirable sequence diversity of receptors and suppression of potentially catastrophic autoimmunity through inadvertent recognition of self.

Here, we propose a solution to this conundrum. As a consequence of TSD during the foundational insertional transposition event interrupting the primordial *VJ* exon, an epistatic entanglement between *V* and *J* sequences could be established by the TSDs serving as microhomologies to guide NHEJ-mediated repair after RSS-directed excision[18,19,21,32,45] (Fig. 4c). Engaging the mechanistic characteristics of NHEJ[46,47] for the *VJ* recombination process therefore minimized the degree of uncertainty arising from junctional diversity. Maintenance of TSD-derived microhomologies to, at least initially, constrain junctional diversity may have provided a critical time window during which to evolve specialized self-tolerance mechanisms as a prerequisite of increasing receptor diversity without the risk of intolerable collateral damage. Moreover, we note that TSD-derived microhomologies not only constrained sequence diversity, but also increased the probability of generating functional (in-frame) recombination products thus supporting efficient lymphocyte development bearing functional receptors. An interesting corollary of this scenario is that the adaptive receptors may have initially functioned more like pattern recognition receptors[48].

If TSDs provided the critical primordial microhomology required for a largely predictable outcome of recombination, it should be possible to trace their modifications over evolutionary time. Indeed, it was possible to identify an evolutionarily ancient microhomology sequence in *TRA*, and to provide evidence for the subsequent degradation of this signature. Our phylogenetic analysis indicates the loss of microhomology signatures in species that show signs of engaging the activity of TdT to increase the diversity of junctional sequences of *TRA* assemblies. When TdT is less active (such as in foetal life)[27–30,32,49] or when TdT is genetically inactivated[34,35], the antigen receptor repertoires show increased reliance on microhomology-based recombination. It is not clear whether the first evolutionary step of increasing junctional diversity was initiated by degradation of microhomology in the absence of TdT activity, or whether the loss of microhomology followed the recruitment of TdT to the assembly process. For instance, some *TRG* assemblies in mice use the same *V* and *J* elements that feature both stereotypic recombination outcomes and extensive junctional diversity, a constellation that favours the retention of microhomologies at the ends of the germline sequences of recombining elements.

It is instructive to compare the scenario that we have proposed for jawed vertebrates to the situation in jawless vertebrates (lampreys and hagfish), which show an alternative adaptive immune system[50]. Here, somatic diversification relies on gene conversion to assemble into arrays different numbers of individual leucine rich repeat-encoding cassettes, guided by identical sequences at their ends, with little if any junctional diversification[51]. Although the repertoire of these leucine rich repeat cassettes measures into the hundreds in extant species, one can easily imagine that at the inception of the mechanism, combinatorial diversity was low and hence the nature of the repertoire of assembled receptors was largely, if not entirely predictable. Thus, in this view, although the main types of assembly process of antigen receptor genes differ between jawless and jawed vertebrates, all vertebrates expanded their recognition capabilities by large-scale gene duplications to increase combinatorial diversity of recombining elements, and further evolved different antigen receptor classes by diversification of constant region sequences. In jawed vertebrates, junctional diversity increased further with the co-option of non-templated DNA polymerases such as TdT into the diversification process, with further options provided by developmental stage-specific regulation of TdT activity.

In summary, our study highlights the evolvability of microhomologies at the ends of *V* and *J* genes in *TRA* loci and suggests a plausible scenario by which a revolutionary but potentially destructive mechanism of antigen receptor diversification could have been successfully integrated into the emerging adaptive immune system of ancient vertebrates.

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

# Methods

## Animals

Zebrafish (*D. rerio*) TU (Tübingen), and TLEK (Tüpfel long fin/Ekkwill) wild-type strains, medaka (*O. latipes*) and mouse strains are maintained in the animal facility of the Max Planck Institute of Immunobiology and Epigenetics, Freiburg, Germany. For zebrafish and medaka, adult fish of both sexes were used; the source of adult *P. progenetica* specimens was previously described[40]. The *Tra*-deficient mouse strain (B6;129S 2-*Tcra^{tm1Mom}*/J)[52] was obtained from The Jackson Laboratory (strain no. 002115); adult mice of both sexes were used. Specimens of unspecified sex from juvenile brown-banded bamboo shark (*C. punctatum*), grey bichir (*P. senegalus*), juvenile sturgeon (*A. ruthenus*), juvenile West African lungfish (*P. annectens*) and adult trout (*O. mykiss*) were obtained from fish dealers. The blood samples from three female adult African bush elephants (Sabie, Tika and Sweni) were obtained from the Wuppertal Zoological Garden and provided by L. Grund. All animal experiments were performed in accordance with relevant guidelines and regulations, approved by the review committee of the Max Planck Institute of Immunobiology and Epigenetics and the Regierungspräsidium Freiburg, Germany (licence AZ 35-9185.81/G-17/79).

## RNA extraction

Animals were euthanized using 0.02% MESAB. Whole fish (zebrafish, medaka), or dissected thymus, spleen and kidney marrow tissues (bamboo shark, bichir, sturgeon, lungfish, trout) were frozen and pulverized in liquid nitrogen, and then dissolved and homogenized in TRIzol reagent (Life Technologies). Mouse lymphocytes were obtained from either the thymus (*Tra*-null mice) or the spleen (wild-type mice); cells were passed through a cell strainer in PBS, centrifuged and the cell pellet dissolved in TRIzol following the recommendations of the manufacturer. For elephant blood samples, mononuclear cells were isolated from roughly 50 ml of peripheral blood as described in ref. 53, using the 1.079 g cm$^{-3}$ Percoll condition; cells were washed and resuspended in TRIzol. Total RNAs were extracted from TRIzol according to the manufacturer's protocol.

## cDNA synthesis

The total amounts of RNA used for cDNA syntheses are recorded in Supplementary Table 5. cDNA synthesis was performed using the SMARTScribe Reverse Transcriptase (Clontech) with an oligo-dT primer (5′-AAGCAGTGGTATCAACGCAGAGTTTTTTTTTTTTTTTTTTTTTTTTTTTTTVN) and SMARTer_Oligo_UMI primer (5′-AAGCAGUGGTAUCAACGCAGAGUNNNNUNNNNUNNNNUCTT[rGrGrGrGrG]) according to the SMARTer RACE 5′RACE protocol (Clontech), using a maximum of 2 µg of total RNA in 40 µl total reaction volume. The SMARTer_Oligo_UMI introduces barcoding at the cDNA level and affords the possibility to enzymatically digest the oligos with uracil-DNA glycosylase. cDNA was purified using the QIAquick PCR Purification Kit (QIAGEN) and eluted in 60 µl of water.

## Amplification of antigen receptor genes

The antigen receptor genes of all species were amplified using the strategy previously described[40], which is a modified version of another previously described procedure[54] (see Supplementary Table 6 for sequence information of primers). The first round of PCR amplification was carried out in a multiplex manner: 1× Q5 buffer, 0.5 mM deoxynucleoside triphosphate, 0.2 µM UPM_S primer (5′-CTAATACGACTCACTATAGGGC), 0.04 µM UPM_L primer (5′-CTAATACGACTCACTATAGGGCAAGCAGTG GTATCAACGCAGAGT) and 0.2 µM of each gene-specific primer (GSP), 15 µl of cDNA, water to 49.5 µl, 0.5 µl of Q5 Hot Start High-Fidelity DNA Polymerase (New England Biolabs); 98 °C for 90 s followed by 20 to 23 cycles of 98 °C for 10 s, 68 °C for 20 s and 72 °C for 45 s, followed by 8-min final extension at 72 °C. GSPs used in the first round are indicated in Supplementary Table 6 with the designation 'outer'. Amplicons

were purified with AMPure XP beads (0.65×) and eluted in 50 µl of water. For the second round of PCR amplification, another multiplex PCR was performed. For each gene, 2% of the first-round amplicon material (1 µl) was used for 25 µl of reactions, using 0.2 µM (combined final concentration) of an equimolar mix of each group of three primers designated 'inner' (Supplementary Table 6). The resulting material was purified with AMPure XP beads (0.65×) and barcoded with NEBNext multiplex oligonucleotides for Illumina by performing four more PCR cycles with 65 °C annealing for 75 s and extension for 75 s, followed by a final extension of 5 min at 65 °C and size selection of amplicons by bead purification as above. Paired-end sequencing runs were performed using a Illumina MiSeq instrument (read length of 300 bp), NovaSeq (read length of 250 bp) or Hiseq (read length of 250 bp) (Supplementary Table 5).

## Generation and analysis of CRISPR mutants

We designed guide RNAs targeting the first exon of the zebrafish TCRα constant region gene (*trac*), situated 5′ of the position of the primers used for amplification of transcripts, using a different set of GSPs (OBG225–OBG228; Supplementary Table 6). This design allows one to distinguish the allelic origin of cDNA molecules; molecules with in-frame stop codons in the *trac* region were classified as 'non-selectable' and analysed separately.

To mutate *Va* genes, guide RNAs were designed to target the most conserved ends of *V* regions in the zebrafish genome. The 3′ ends of the *V* nucleotide sequence until the heptamer corresponds to TGTGCTC**TG** AG**G**CC, with the TGT triplet coding for the characteristic cysteine residue. The PAM site (underlined) partially overlaps with the residues used for microhomology-guided repair (bold face); hence CRISPR–Cas9-mediated mutations were expected to displace them together with the RSS sequence, producing frameshift in the assembled CDR3 sequences (relative to the wild-type situation), whenever the number of insertions and/or deletions was not a multiple of three. The resulting CDR3 sequences were scanned for the last six nucleotides of our guide sequence, and split into sequences containing them at the usual position (control) or displaced by one nucleotide (mutant).

We followed the methods previously described[55] for the generation, testing and general injection methodology. The target sequences for the mutagenesis experiments are as follows. *trac* mutation 5′-AAG CCGAATATTTACCAAG; *Va* mutation 5′-CTGTGTATTACTGTGCTCTG.

## Reference genomes

For repertoire and phylogenetic analyses, genome assemblies were obtained from publicly available sources: National Center for Biotechnology Information (NCBI) (https://www.ncbi.nlm.nih.gov/genome/), Ensembl (https://www.ensembl.org/index.html) and Squalomix (https://transcriptome.riken.jp/squalomix/). For *tra* and *trb*, the *V*, *D*, *J* and *C* elements were identified (Supplementary Tables 2 and 3); when no complete genome assembly was available, relevant scaffolds were concatenated without regard to their true order; this does not affect the analysis, because each element is considered here as a separate entity. For lungfish, only one of the two *tra* loci was analysed.

## Identification of immune gene elements, estimation of lymphocyte count

Our analysis was started by in-depth analysis of the immune gene constellations in zebrafish and mouse, using the IMGT (ImMunoGeneTics) database https://www.imgt.org/ as initial reference. Gene segments were mapped by sequence identity to danRer11 (UCSC, release date May 2017) and mm10 (UCSC, release date September 2017) genome assemblies, and informatically analysed using tools developed relying on the R BSgenome package[56]. The zebrafish *tra* and *trb* loci have been previously described[57,58]; during the course of this work, we identified four previously unrecognized *Va* elements, and 14 previously unrecognized *Ja* elements that map to the genome and form canonical

rearrangements. An adult zebrafish harbours between 200,000 and 300,000 T cells[59–61]. The *tra* locus in trout has been recently described[62]; the *TRA* loci of other species were identified and characterized in this work (below).

## Identification of *tra* and *trd* constant region genes in genome assemblies

The TCR constant region genes were identified by sequence similarity to closely related species. We used published data[63] to identify peptide signatures of *trac* and *trdc* exon 1 sequences (*tra* CLXTD followed by F or XF; *trd* CLXXXFXP; X stands for any amino acid residue). The correct designation of these two constant regions was subsequently confirmed by the identification of clusters of *Ja* elements (below) in the canonical 5′-*trdc*–(*traj*)$_n$–*trac*-3′ configuration.

## Identification of *Ja* genes in genome assemblies

To identify *Ja* clusters in genomes for which we had no repertoire data available as an independent reference, we used a method based on sequence similarity. We found that for all the species used in the repertoire analysis, the distance from (and including) the characteristic FGXG tetrad of *Ja* sequences to the intron donor site was 34 nucleotides (Extended Data Fig. 2). By aligning the nucleotide sequences of Jα elements of three teleost species (*P. progenetica*; *D. rerio*; *O. mykiss*) and two mammalian species (*M. musculus*; *L. africana*), and using 0.6 bits of entropy as a maximum threshold per position, we obtained the following pattern, ending in the intron donor (*gt*): TN$_4$TTNGGN$_4$GGNAC-N$_5$TN$_5$N$_8$*gt*, in which N is any letter in the International Union of Pure and Applied Chemistry code. This pattern is expected to happen by chance once every $2^{26}$ (roughly 67,000,000) nucleotides, whereas the length of a typical *Ja* region is in the order of 50,000 to 200,000 nucleotides. In addition to the nucleotide pattern for identification, we also used the FGXGTX[LV]X[VI] canonical pattern as a search sequence, and constrained the search by the canonical 5′-*trdc*–(*traj*)$_n$–*trac*-3′ configuration. Rare unconventional *Ja*-like sequences presenting with a variant tetrad (such as FAKG) were not included in this part of the analysis as such elements might also be present in species that we did not evaluate by repertoire analysis and hence have no means to ascertain their apparent functionality. The search algorithm described above detects on average around 80% (range 67.1 to 89.6%) of the *Ja* elements that were found in the sequenced repertoires of the species, which were not used to generate the nucleotide search pattern (*C. punctatum*, *P. senegalus*, *A. ruthenus*, *O. latipes*, *P. annectens*).

## Identification of RSS in genome assemblies

The positions of RSS sequences of *Va* and *Ja* elements[64] were identified by use of known RSS sequences of zebrafish and mouse. A matrix with the nucleotide frequencies in these RSS sequences was used as input; a score for each nucleotide was generated using the PWMscoreStartingAt function of the R Biostrings package[65]. The highest score for each sequence was chosen as the RSS position. From the newly identified RSS sequences, a new matrix was generated, and the process repeated through five cycles. The results of these algorithms converge when starting with either zebrafish or mouse RSS matrices as query (Extended Data Fig. 9). Note that RSS positions are evaluated only after *Ja* elements had been identified by the similarity patterns described in the section Identification of *Ja* genes in genome assemblies. As the RSS is typically located some 20 nucleotides 5′ of the query pattern used for the identification of *Ja* elements, and hence does not include the FGXG signature, the subsequent RSS identification is unlikely to be biased by the outcome of the initial *Ja* identification.

## Immune repertoire data extraction

To extract *V* and *J* sequences from amplified *TRA* and *TRB* assemblies, we expanded on our previous R pipeline available at GitHub (https://github.com/obgiorgetti/minifish). The code for the current version (https://github.com/obgiorgetti/TCRalpha) follows the same method. In a first step, unique molecular identifier (UMI) barcodes were matched to CDR3 regions (including the entire *J* sequence), followed by *V* gene sequence identification. Each unique combination of UMI, *V*, CDR3 and *J* sequences was considered to represent a single cDNA molecule; however, it was kept for analysis only if it was read more often than a certain threshold (Supplementary Table 5) and was otherwise discarded. Then, we performed two levels of error corrections on the basis of UMIs (Supplementary Table 5). (1) Sequences of the same CDR3 length, where UMIs are at a Hamming distance of one nucleotide, and CDR3 sequences are at a Hamming distance of two nucleotides or less were considered errors, as UMI and CDR3 sequences should be independent; in each of such instances, from the graph that connects all such neighbouring UMI + CDR3 sequences, we retained the variant with highest numbers of reads. (2) A subsequent error correction was carried out for UMIs that, after the first correction, are associated with two or more CDR3s. In these situations, we kept sequences at a Levenshtein distance greater than three (or the most read sequence in case of conflict). This correction removes errors created by nucleotide insertions, which although less frequent than substitutions, occur particularly in CDR3s with long strings of repeated nucleotides. For the species in which we obtained full repertoire data, the mapping of *V* segments was done with the 3′ read of the paired reads; it proved difficult to consistently map the 5′ ends in non-model species due to the pervasive presence of single-nucleotide polymorphisms and likely inaccuracies in the available assemblies. On the basis of the repertoire data, we constructed a table of expressed *V* segments for each species, and mapped each to the available genomes (Supplementary Tables 2 and 3). This table was constructed in the following way. We started by identifying the constant region in the cDNA sequences using the signature described above. Then, open reading frames (ORFs) of at least 60 amino acid residues in lengths were extracted (using UMIs to remove sequencing mistakes); the generic signature of J elements (FGXGTKL or its close variations) were used to define the correct ORF. In these ORFs, we searched for a cysteine residue (allowing a distance of up to 20 amino acids upstream of the phenylalanine residue in the *J* element). The positions of the cysteine residues identified in this manner were used as reference points to extract 180 nucleotides of *V* elements from the cDNA sequences; this collection constitutes the dictionary of expressed *V* elements, which is subsequently mapped to the germline *V* dictionary, allowing up to 5 nt distance. Once the *V* elements were identified, it was possible to delimit the lengths of CDR3 regions by comparing the cDNA sequences against those of *J* regions. For this, a list of *V* and *J* polymorphisms was composed to correctly identify and map the *V* and *J* nucleotides in CDR3 sequences. We determined the presence of single-nucleotide polymorphisms in a stretch of 15 nucleotides of germline sequences directly adjacent to the RSS at the 3′ ends of *V* elements, or at the 5′ ends of *J* elements, respectively. *V* and *J* elements in the expressed repertoire that are not found in the available genome assemblies were excluded in the analysis, as it is not possible to unambiguously assign the position of RSS elements relative to their reading frames. For our repertoire pipeline, we used a *V* dictionary and a *J* dictionary for germline assignment, and used the germline sequences of these two segments to delimitate the CDR3 an end of *V* consensus amino acid pattern and *J* consensus amino acid pattern.

To exclude the possibility that the process of non-sense mediated decay of mRNAs interferes with the analysis of *VJ* assemblies transcribed from the mutant *tra* allele of zebrafish, we determined the number of UMIs as a representative of the number of mRNA molecules. We found that for heterozygous fish, roughly 48% of molecules in the repertoire originated from the wild-type allele and roughly 52% from the mutant allele, suggesting that non-productive *tra* mRNAs do not undergo non-sense mediated decay.

For the analysis of *TRG* and *IGL* loci (Supplementary Figs. 1–5), IMGT reference genes (https://www.imgt.org/) were mapped to the same

genome assemblies that were used for the *TRA* and *TRB* loci. In the case of *TRG* of *D. rerio*, for which no such reference database for *V* and *J* elements could be found, 64 assembled sequences deposited in the GenBank database (accession numbers AY973880.1 to AY973943.1) were used for the mapping the *TRG* locus. The corresponding genomic coordinates (*D. rerio*; GCA_000002035.4; NCBI; all on a minus strand) are as follows: TRGC1 (34856954-34856986); TRGJ7 (34861351-34861530; RSS at −47); TRGJ6 (34861917-34862096; RSS at −58); TRGJ5 (34862567-34862746; RSS at −52); TRGJ4 (34863715-34863894; RSS at −52); TRGJ3 (34864064-34864243; RSS at −49); TRGJ2 (34864397-34864576; RSS at −55); TRGJ1 (34865455-34865634; RSS at −48); TRGV7 (34866745;34866924; RSS at +27); TRGV6 (34869141-34869320; RSS at 24); TRGV5 (34873039-34873218; RSS at 22); TRGV4 (34877490;34877669; RSS at 22); TRGV3 (34880215-34880394; RSS at 25); TRGV2 (34885611-34885790; RSS at 34); TRGV1 (34888996;34889175; RSS at 25.

For the analysis of the *TRD* locus of *P. progenetica* (Supplementary Fig. 6), the data were taken from Giorgetti et al.[40].

## Phylogenetic analysis

We constructed a phylogenetic tree derived from the Open Tree of Life using the rotl R package[66,67]. Tree tip aesthetics were modified using ape[68] and phanghorn[69] packages. The sequence sources for the analysis of *Ja* elements in vertebrate genomes are listed in Supplementary Table 4.

## Entropy analysis

Previous methods aimed at estimating the entropy of immune receptor repertoires focused on a mathematical description of the *V*(D)*J* recombination process[70]. In the present work, we were confronted with the challenge of comparing antigen receptor repertoires potentially arising from different generative strategies. Thus, our main focus was to be able to identify the germline-encoded segments in CDR3 regions. To account for the non-independence of nucleotides in codon triplets, we also calculate the conditional entropy of amino acid residues in CDR3 regions.

Given the random variables: *S*, full sequence of TCR; CDR3, sequence in either nucleotide or amino acid, covering the segment corresponding to the conserved cysteine and phenylalanine/tryptophan residues; *V* denotes *V* gene; *J* denotes *J* gene; and *L* denotes CDR3 length in nucleotides or amino acid residues, we want to estimate the entropy *H* of *S*:

$$H(S) = H(\text{CDR3}, V, J) = H(\text{CDR3}|V,J) + H(V,J)$$

We start by separating CDR3s by length, and estimate for each length *l* in *L* the entropy using the measured frequencies of each variable:

$$H(S|L=l) = H(\text{CDR3}|V,J,L=l) + H(V,J|L=l)$$
$$= H(\text{CDR3}|L=l) - I(\text{CDR3};V,J|L=l) + H(V,J|L=l)$$

*H*(CDR3|*L* = *l*) is a shorthand for *H*(CDR3*n*|*L* = *l*), which is simply the entropy of each position *n* given a length *l*, with a maximum of two bits, and corresponds to the bar height in our graphic depiction (Fig. 1d and Extended Data Fig. 8), while *I*(CDR3; *V*,*J*|*L* = *l*) is the mutual information between each CDR3 position and *VJ* pairs, therefore with a maximum of H(CDR3*n*|*L* = *l*) bits.

We avoid using *VJ* pairs and take the maximum values of CDR3 and either *V* or *J* separately:

$$\max(I(\text{CDR3};V|L=l),\ I(\text{CDR3};J|L=l))$$

and this later version is depicted in blue (if *V* was used) and red (if *J* was used). *V* and *J* have low mutual information content, therefore are essentially independent.

With libraries that are deeply sequenced as to give an accurate representation of the CDR3 composition of each *VJ* pair for every CDR3

length, the mutual information could be calculated with the formula presented above instead and would be expected to yield a slightly higher value, therefore decreasing the final entropy estimate. Note that in this case, alphabet size would be $L \times V \times J \times 4$, whereas with our simplification it is $L \times V \times 4$ or $L \times J \times 4$, and this is why that method would require deeper sequencing.

The weighted sum of the formula above over all *l* in *L* gives

$$H(S|L) = \sum l \in L \mathrm{p}(l) H(S|L=l)$$

and last from Bayes' rule for conditional entropy we obtain:

$$H(S) + H(L|S) = H(L) + H(S|L)$$

$$H(S) = H(L) + H(S|L) - H(L|S)$$

$$H(S) = H(L) + H(S|L)$$

where the *H*(*L*|*S*) is 0, because if the sequence is known, then its length is also known.

Therefore, we use the weighted sum of the conditional entropy given the length plus the entropy of the length distribution to estimate sequence entropy.

## Reporting summary

Further information on research design is available in the Nature Portfolio Reporting Summary linked to this article.

## Data availability

The GenBank accession numbers for primary data are PRJNA865512 and PRJNA865921. All other data are available in the text or in supplementary materials. Source data are provided with this paper.

## Code availability

The codes used for the analyses in this paper are available at github.com/obgiorgetti/TCRalpha.

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

**Acknowledgements** We thank L. Grund for provision of elephant tissues and D. Schatz for insightful comments on an earlier version of the manuscript. This work was supported by the Max Planck Society.

**Author contributions** O.B.G. and T.B. conceptualized and planned the project. O.B.G. performed the repertoire sequencing and developed the computational strategy. O.B.G. and C.P.O'M. conducted the mutagenesis experiments. M.S. assisted with animal handling and tissue procurement. All authors analysed results. O.B.G. and T.B. wrote the manuscript, with input from all authors. T.B. conceived and supervised the project.

**Funding** Open access funding provided by Max Planck Society.

**Competing interests** The authors declare no competing interests.

**Additional information**
**Correspondence and requests for materials** should be addressed to Orlando B. Giorgetti or Thomas Boehm.

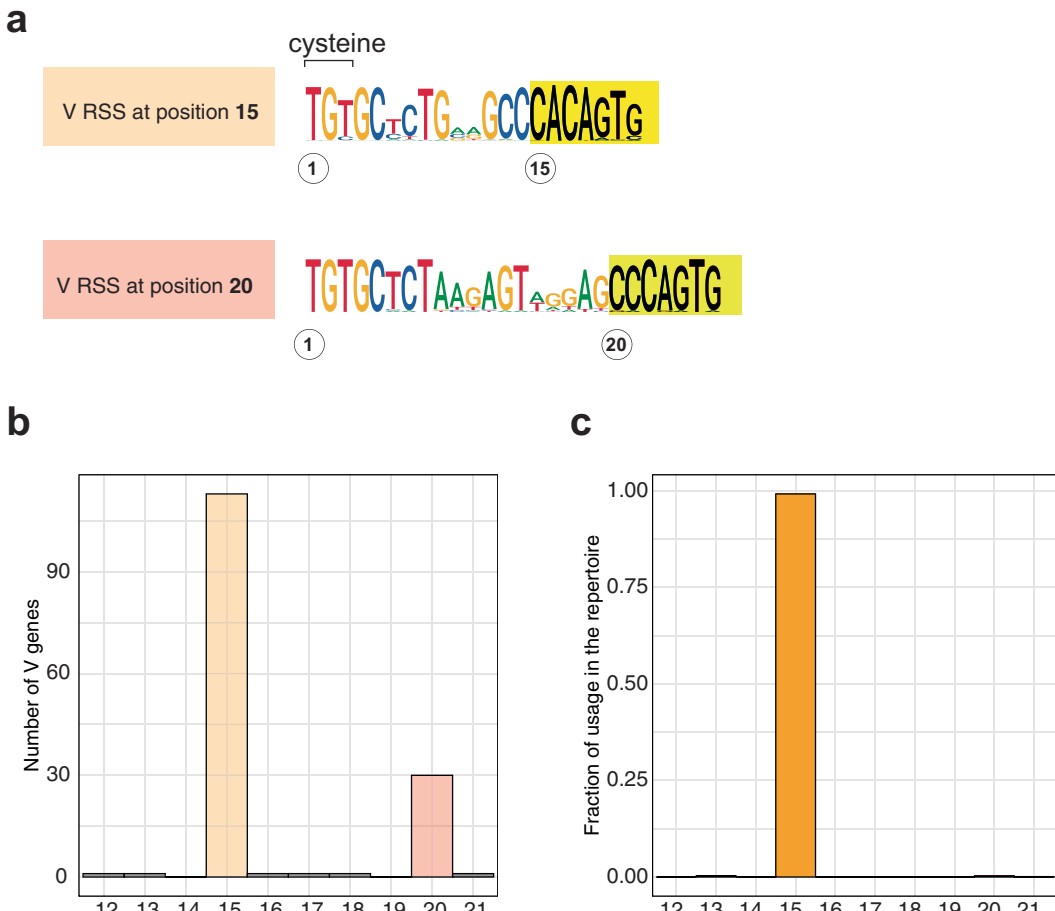

**Extended Data Fig. 1 | Position of RSS elements in *tra* variable genes of *Danio rerio*. a**, Two major classes of *V* elements are present in the genome. When counting the first nucleotide of the conserved cysteine codon at the 3′-end of *V* elements as nucleotide +1, the first nucleotide of the RSS element (only heptamer sequence is shown) occurs either at position 15 or 20, respectively. The non-canonical CCCAGTG heptamer is an unfavourable element (see methods), but is used, albeit rarely, in *VJ* assemblies. **b**, Two major size classes of *V* elements grouped according to the position of the RSS in the genome. **c**, Almost all *V* elements expressed as *tra* assemblies are those which exhibit the RSS element at position 15.

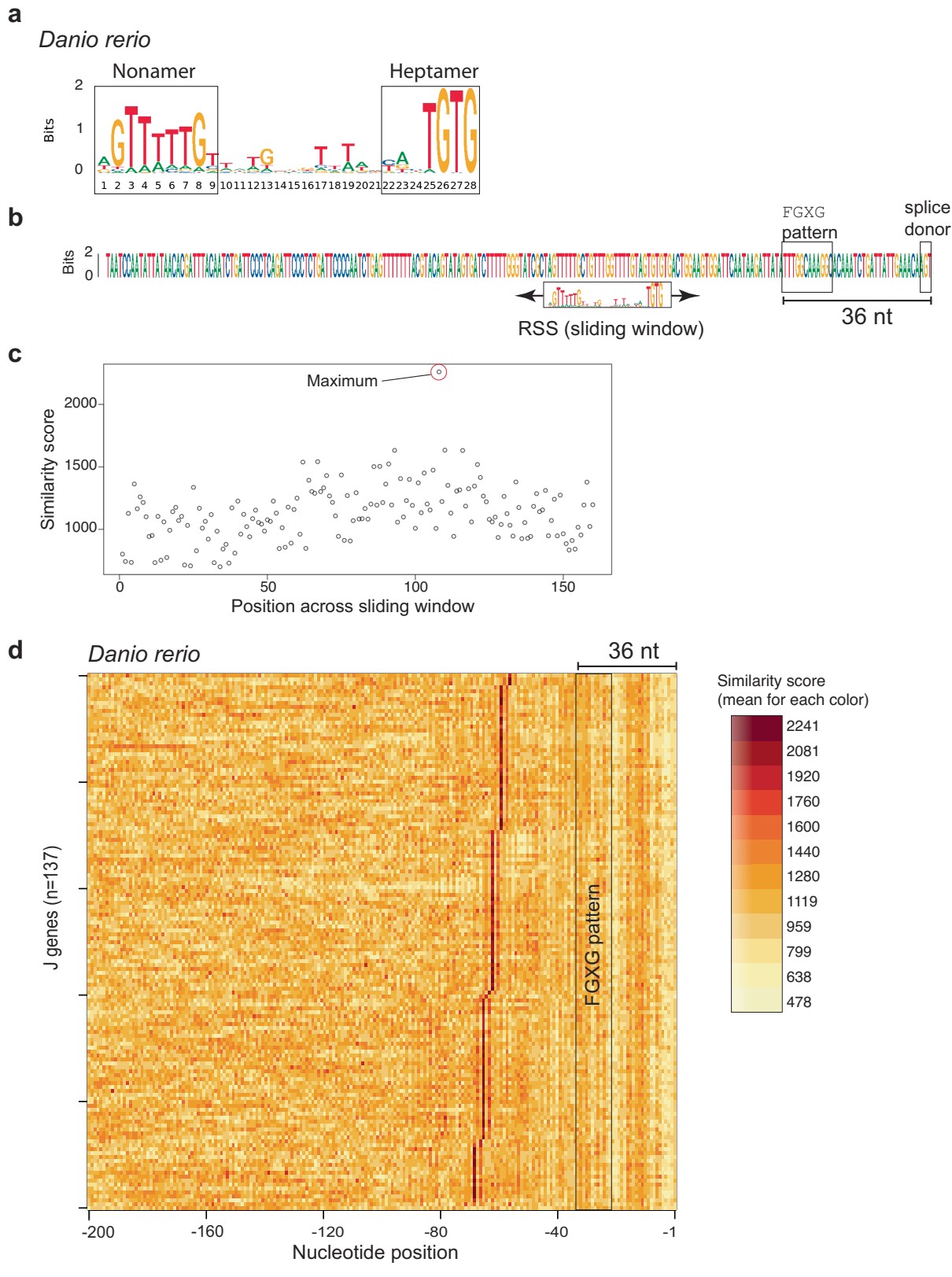

**Extended Data Fig. 2 | Strategy to identify the RSS elements of *Ja* genes in assembled genomes. a**, A consensus sequence was derived for *Ja* RSS elements of *D. rerio* and other species by manual curation. **b**, The RSS consensus sequence was used as a search pattern against the nucleotide sequence of genomic *Ja* regions; an example containing a *Ja* element is shown. Position −1 corresponds to the thymidine residue in the canonical GT splice donor site. **c**, The consensus was compared against the genomic sequence in steps of 1 nucleotide and at each step a similarity score was calculated using the PWM function of the R Biostrings package. For the example shown in (b), the position of the RSS is assigned to a unique position (red circle). **d**, Identification of RSS positions as expressed in similarity scores (colour-coded) across a 200-nucleotide window upstream of the splice donor sites of *J* genes of *D. rerio*. Numbering of nucleotide positions as in (b).

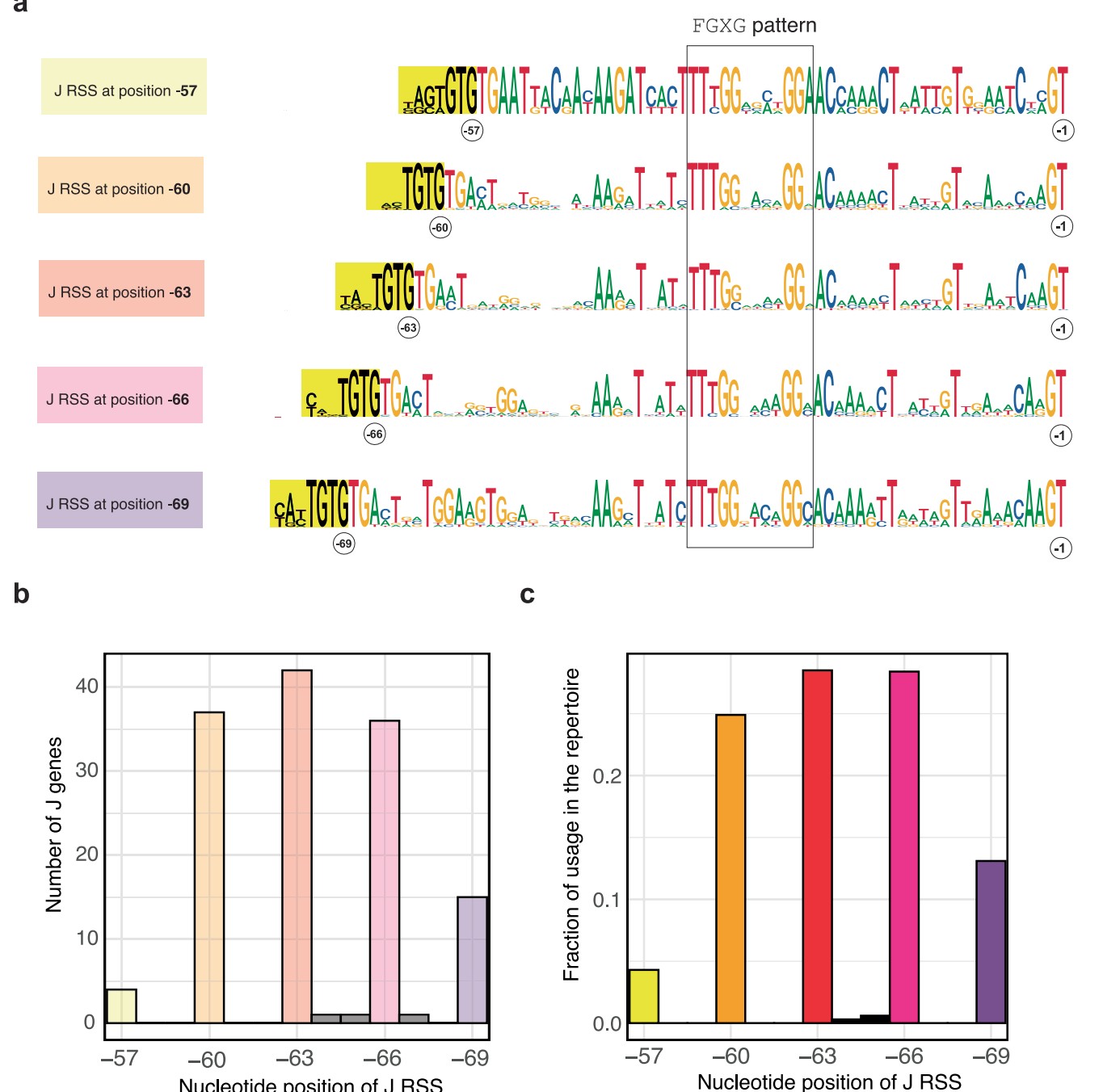

**Extended Data Fig. 3 | Position of RSS elements in *traj* genes of *D. rerio*.**
**a**, Five major classes of *J* elements are present in the genome, distinguished by the distance between the RSS element (only heptamer sequence is shown) and the conserved Phe-Gly-X-Gly peptide tetrad. The distance from the FGXG to the splice donor site at the 3′-end of *J* elements is always identical. The numbering scheme follows Extended Data Fig. 2. **b**, Number of *J* elements in the genome, grouped relative to the position of RSS. **c**, Size distribution of expressed *J* elements, grouped relative to the position of RSS.

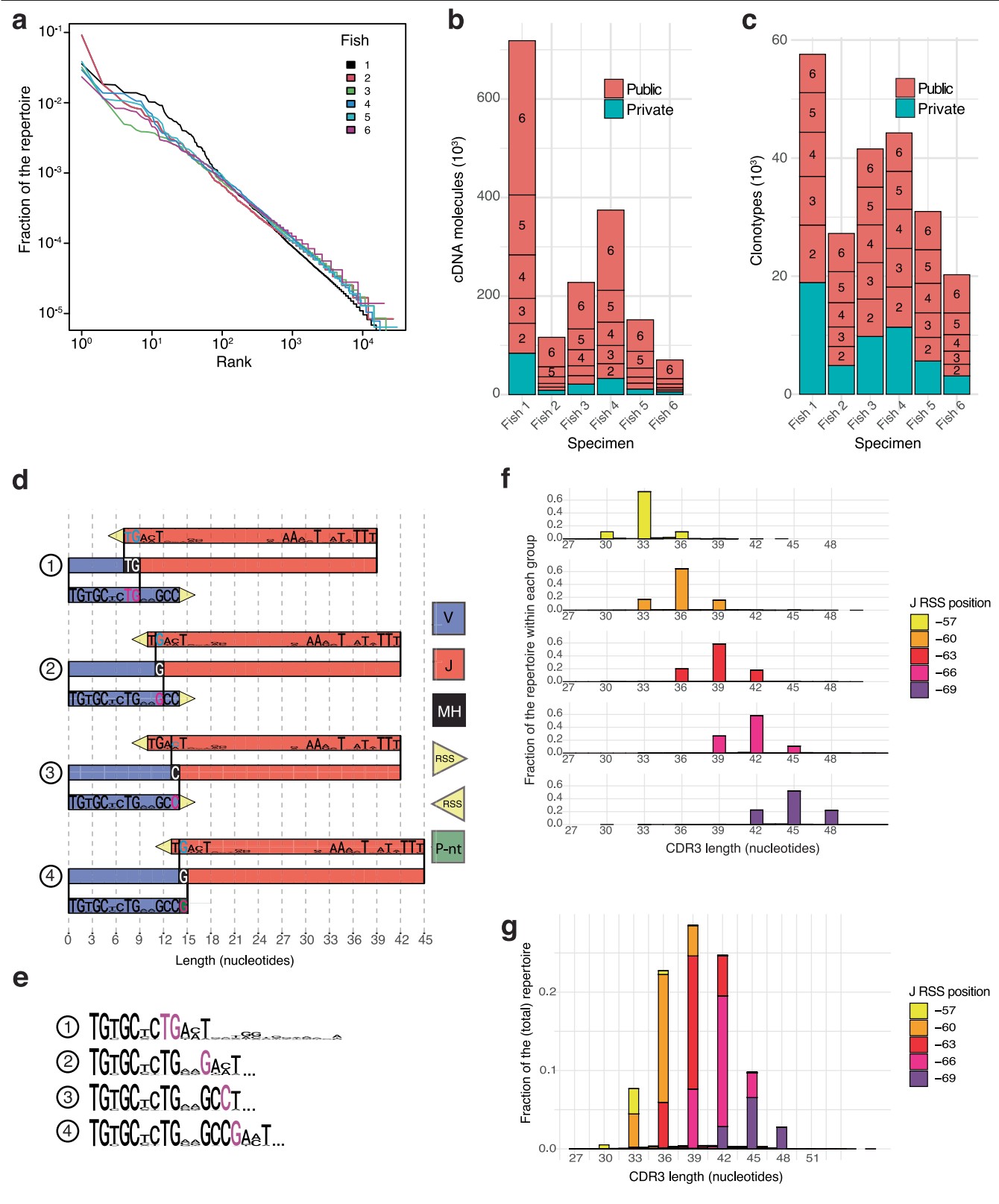

**Extended Data Fig. 4 |** See next page for caption.

**Extended Data Fig. 4 | Microhomology-directed recombination of *tra* of *D. rerio*. a**, Fraction of UMI counts of individual clonotypes ranked by their frequencies for 6 zebrafish; these distributions follow a power law indicative of the fractal nature of the repertoires. **b**, Overlap of sequences among the repertoires of 6 fish as determined on the basis of UMI counts. The numbers in boxes indicate the overlaps among the numbers of fish. Note that the public sequences present in all fish make up the largest parts of the repertoires. **c**, Overlap of sequences among the repertoires of 6 fish as determined on the basis of clonotypes. **d**, Schematic representation of the four major microhomology patterns. Indicated are the 5′- ends of germline sequences of *J* elements (top; the last three nucleotides correspond to the phenylalanine residue in the FGXG tetrad); the 3′-ends of germline sequences of *V* elements (bottom; the first three nucleotides correspond to the cysteine residue at the C-terminus of *V* elements); and the resulting assembled sequence (middle). The presumptive contributors to microhomology-directed repair are coloured; in pattern 4, the G residue in the *V* sequence represents a P nucleotide. **e**, Consensus sequences of assemblies arising from the four major recombination patterns; shown are nucleotide sequences starting from the cysteine codon in *V* elements to the conserved thymidine residue at the beginning of *J* elements; residues corresponding to the overlap between *V* and *J* genes are coloured. **f**, Length distributions are shown for *J* elements corresponding to those of the five most prevalent RSS positions (see Extended Data Fig. 2). **g**, The total CDR3 length distributions in a repertoire emerges as a superposition of individual distributions as depicted in **f**. The numbering of RSS positions in *J* elements follows Extended Data Fig. 2.

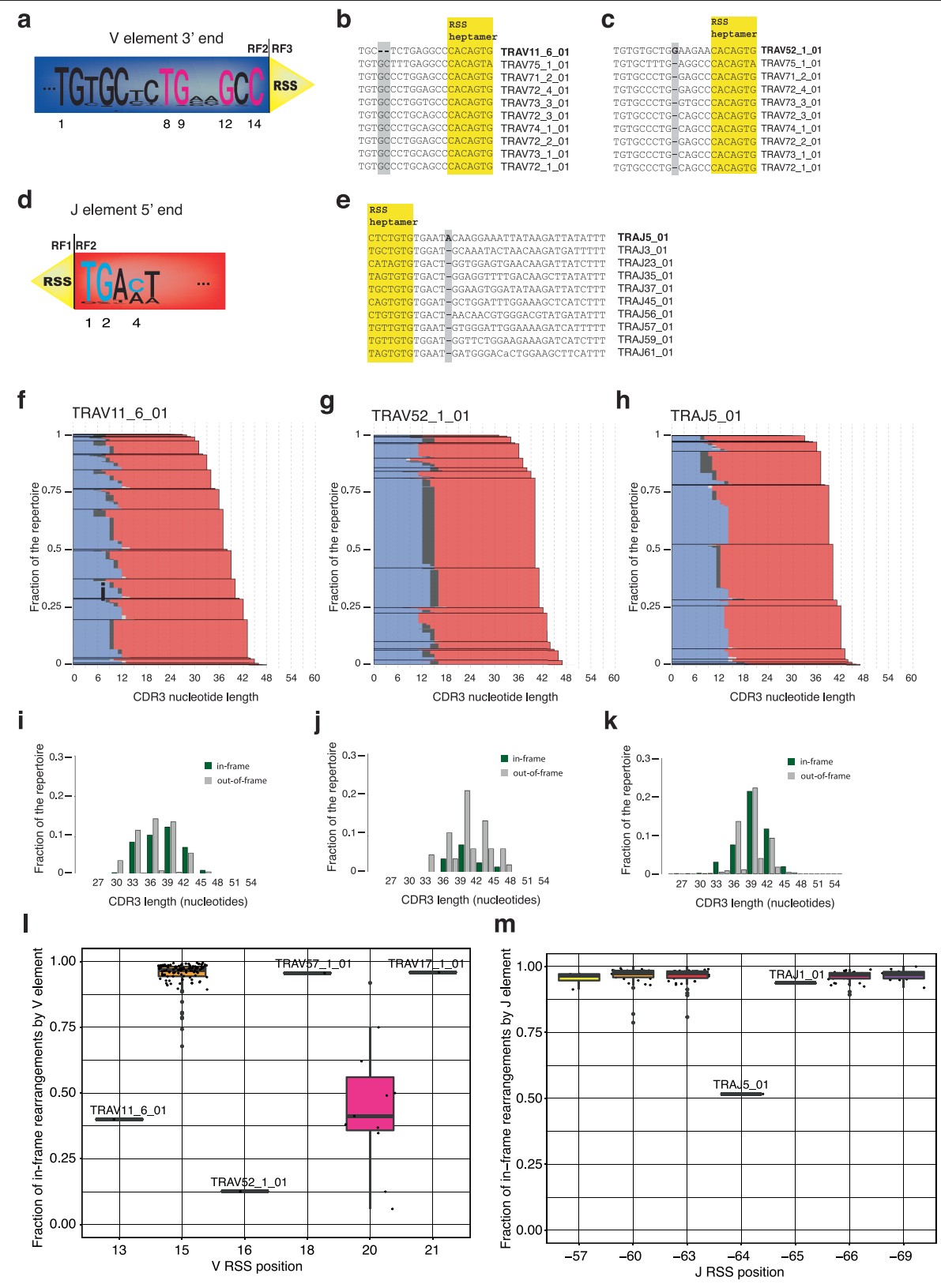

**Extended Data Fig. 5** | See next page for caption.

**Extended Data Fig. 5 | Stereotypic *tra* assemblies of germline *V* and *J* elements of *D. rerio* with unusual sequence compositions. a**, Consensus sequence of the 3′-ends of *V* elements, with important microhomology sites coloured. The RSS elements are inserted after the second nucleotide in the reading frame of *V* genes, at position 15, when counting from the first nucleotide of the conserved cysteine codon. **b**, Sequence of the *TRAV11_6* gene (top), where RSS has been displaced by the deletion of two nucleotides (*V* RSS position 13), compared to the sequences of 9 other *V* genes where RSSs are in a typical frame (*V* RSS position 15). **c**, Sequence of the *TRAV52_1* gene (top), in which the position of the RSS element is displaced by the insertion of one nucleotide (*V* RSS position 16), compared to sequences of 9 other *V* genes where RSSs are in a typical frame (*V* RSS position 15). **d**, Schematic representation of 5′ ends of *J* genes, with important microhomology sites coloured. The RSS elements positioned between the first and second codon of the reading frame. **e**, Sequence of the *TRAJ_5* gene (top) where the RSS has been displaced by the insertion of one nucleotide (RSS now at position -64 relative to the splice donor site); the sequences of 9 other *J* genes with typical position of RSS (RSS at position -63) are also shown. **f**, Repertoire representation for rearrangements utilizing the anomalous element *TRAV11_6*, indicating the presence of microhomology-directed recombination (black segments). **g**, Repertoire representation for rearrangements utilizing the anomalous element *TRAV52_01* element, indicating the presence of microhomology-directed recombination (black segments). Out-of-frame sequences dominate the CDR3 length distribution in joints utilizing *TRAV11_6*. **h**, Repertoire representation for rearrangements utilizing the anomalous element *TRAJ5_01*, indicating the presence of microhomology-directed recombination (black segments). **i**, Out-of-frame sequences dominate the CDR3 length distribution in joints utilizing *TRAV11_6*. **j**, Out-of-frame sequences dominate the CDR3 length distribution in joints utilizing *TRAV52_01*. **k**, Out-of-frame sequences dominate the CDR3 length distribution in joints utilizing *TRAJ5_01*. **l**, Average fraction of in-frame assemblies for *V* genes with different RSS positions when counting from the first nucleotide of the conserved cysteine codon (see Extended Data Fig. 1). *V* elements exhibiting unusual positions are distinguished by a low frequency of in-frame assemblies. **m**, Average fraction of in-frame assemblies for *J* genes with different RSS positions relative to the splice donor site (see Extended Data Fig. 2). Note the low frequency of in-frame assemblies for the rogue *TRAJ5_01* element.

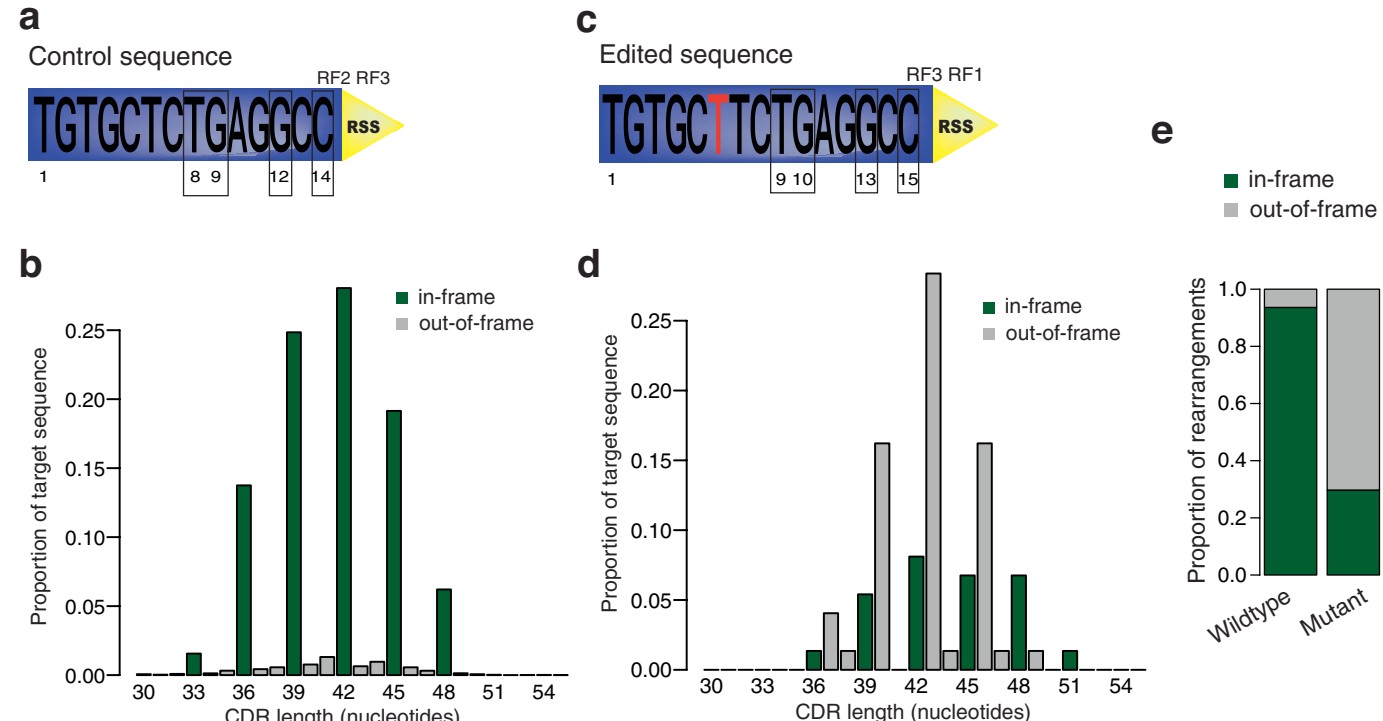

**Extended Data Fig. 6 | Site-directed mutagenesis of *V* elements confirms the dominance of microhomology-mediated recombination. a**, Consensus logo representation for control *V* sequences that exhibit the wild-type target sequence; evolutionarily conserved microhomology positions are highlighted (boxes). **b**, CDR3 length distribution of CDR3s of assemblies utilizing the *V* elements containing the consensus shown in **a**. **c**, Consensus logo representation of mutated *V* sequences corresponding to those shown in **a**, exhibiting an insertion of one nucleotide; note the displacement of evolutionarily conserved microhomology positions (boxes) and, consequently, the reading frame relative to the position of the RSS. **d**, CDR3 length distribution (nucleotides) of CDR3 sequences for assemblies utilizing the mutated *V* sequences. **e**, Proportions of in-frame rearrangements in assemblies using wild-type and mutant *V* elements.

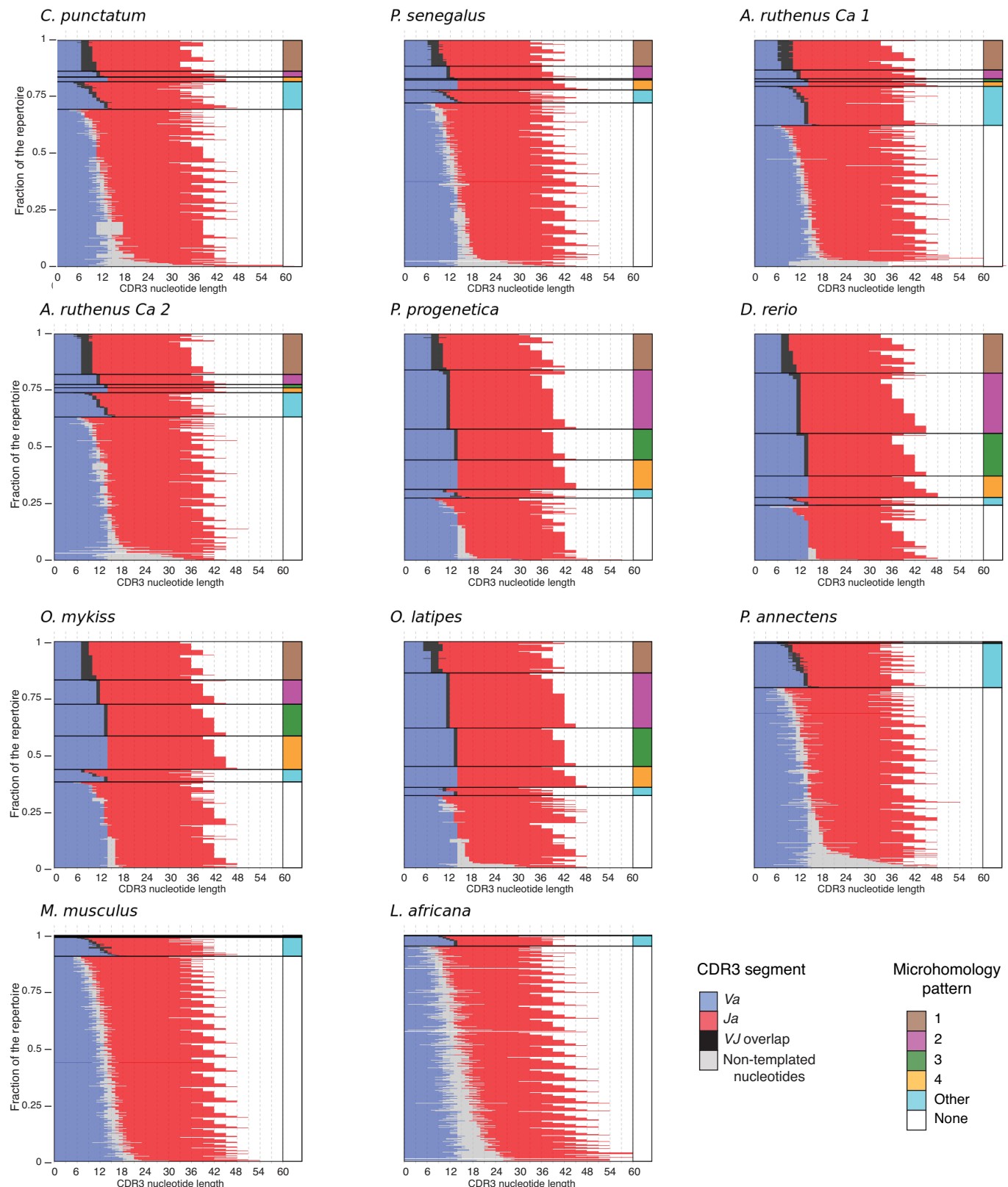

**Extended Data Fig. 7 | Contributions of the four dominant microhomology patterns defined in *D. rerio* to *tra* assemblies in other vertebrate species.** The combined contributions of other types of microhomology-directed recombinations are also indicated.

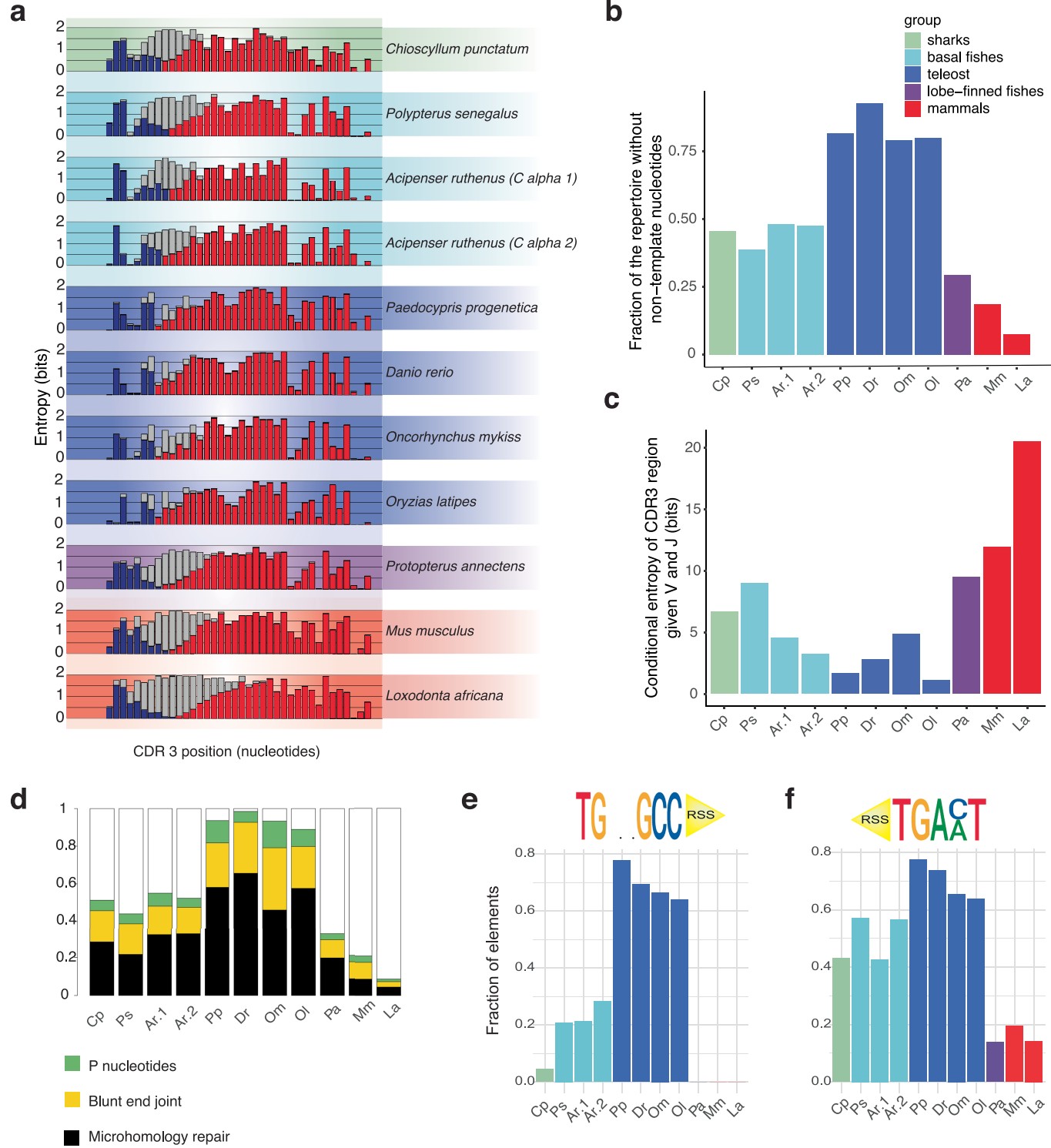

**Extended Data Fig. 8 | Evolutionarily variable contributions of *V* and *J* elements to the entropy of *TRA* CDR3 sequences. a**, Entropy values for nucleotide positions for CDR3 regions of 42 nucleotides in length for 10 species. Indicated are germline-dependent contributions to mutual information between nucleotide and gene (*V* gene, blue bars; *J* gene, red bars) and germline-independent contributions (conditional entropy when *V* or *J* genes are known, grey bars). The species names are given to the right. Note that *A. ruthenus* possesses two *tra* loci. **b**, Fraction of *tra* assemblies without non-templated nucleotides in the repertoires of 10 species. **c**, Weighted average of entropies that are independent of *V* and *J* elements across all CDR3 lengths. **d**, Fractions of CDR3 sequences that can be explained without invoking the activity of TdT according to the three proposed generative mechanisms (green, P-nucleotide based microhomology-guided repair; yellow, blunt-end joining; black, generated by germline microhomology-guided repair). **e**, Presence of the indicated *Va* consensus sequence (top) in the *Va* elements of different species. **f**, Presence of the indicated *Ja* consensus sequence (top) in the *Ja* elements of different species. For b - f, species designations are: *Cp*, *Chioscyllum punctatum*; *Ps*, *Polypterus senegalus*; *Ar*.1, *Acipenser ruthenus* (TCRα locus 1); *Ar*.2, *Acipenser ruthenus* (TCRα locus 2); *Pp*, *Paedocypris progenetica*; *Dr*, *Danio rerio*; *Om*, *Oncorhynchus mykiss*; *Ol*, *Oryzias latipes*; *Pa*, *Protopterus annectens*; *Mm*, *Mus musculus*; *La*, *Loxodonta africana*. For **b**,**c**,**e**,**f**, the colours of bars correspond to the species groups indicated in **b**.

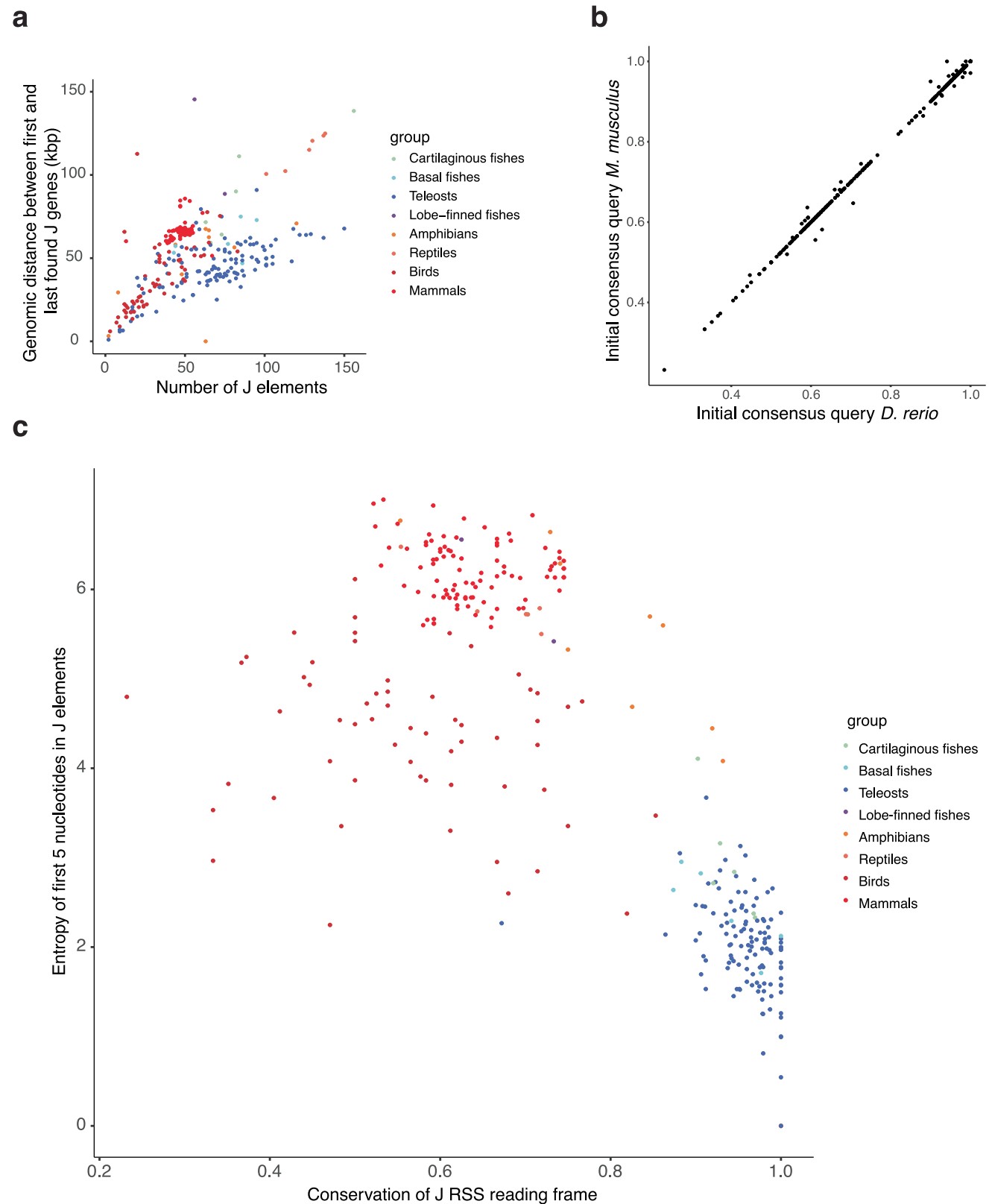

**Extended Data Fig. 9 | Characteristics of *Ja* regions and evolutionary variability of the 5′-ends of *Ja* elements in assembled genomes of vertebrates. a**, The size range of the genomic regions containing *Ja* elements is compared to the number of identifiable *Ja* elements. The data point corresponding to *Catharus ustulatus* is omitted; its *Ja* region is 737,550 bp long and contains 67 *Ja* elements. **b**, The successful identification of RSS positions in genomic *Ja* sequences of vertebrate species (n = 302) is independent of the origin of RSS consensus sequences initially used for the iterative search process, as indicated by the correlation of search outcomes starting with mouse and zebrafish sequences, respectively. **c**, In species with a fixed position of RSS elements relative to the reading frame of *Ja* elements, the entropy of the sequence at the end of *J* elements is low; by contrast, non-conserved positions of RSS elements are associated with high entropy and hence a high degree of sequence variability.

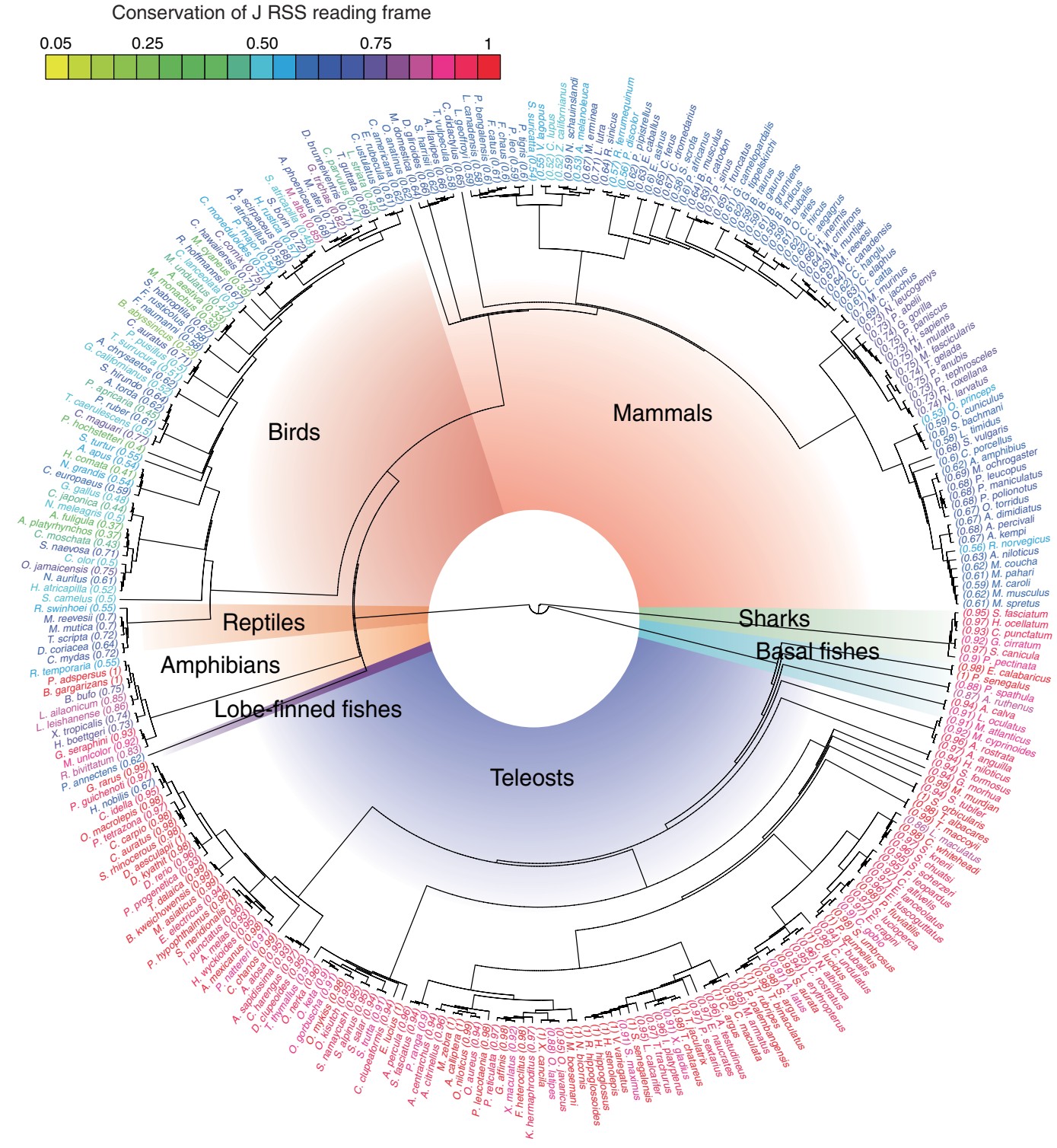

**Extended Data Fig. 10 | Structure of *Ja* regions in vertebrate species.**
A cladogram of 292 out of the 302 species analyzed is shown. Species names
are coloured according to exhibiting the primordial positions of the RSS relative
to the reading frame of the *Ja* gene as depicted in Fig. 2b; this proportion is also
shown in parenthesis. Note that 10 species listed in Supplementary Table 4 are
excluded from the cladogram, because they are not incorporated in the Open
Tree of Life database.

# Reporting Summary

## Statistics

For all statistical analyses, confirm that the following items are present in the figure legend, table legend, main text, or Methods section.

| n/a | Confirmed | |
|---|---|---|
| ☐ | ☒ | The exact sample size (*n*) for each experimental group/condition, given as a discrete number and unit of measurement |
| ☐ | ☒ | A statement on whether measurements were taken from distinct samples or whether the same sample was measured repeatedly |
| ☐ | ☒ | The statistical test(s) used AND whether they are one- or two-sided *Only common tests should be described solely by name; describe more complex techniques in the Methods section.* |
| ☒ | ☐ | A description of all covariates tested |
| ☐ | ☒ | A description of any assumptions or corrections, such as tests of normality and adjustment for multiple comparisons |
| ☐ | ☒ | A full description of the statistical parameters including central tendency (e.g. means) or other basic estimates (e.g. regression coefficient) AND variation (e.g. standard deviation) or associated estimates of uncertainty (e.g. confidence intervals) |
| ☐ | ☒ | For null hypothesis testing, the test statistic (e.g. *F*, *t*, *r*) with confidence intervals, effect sizes, degrees of freedom and *P* value noted *Give P values as exact values whenever suitable.* |
| ☒ | ☐ | For Bayesian analysis, information on the choice of priors and Markov chain Monte Carlo settings |
| ☒ | ☐ | For hierarchical and complex designs, identification of the appropriate level for tests and full reporting of outcomes |
| ☒ | ☐ | Estimates of effect sizes (e.g. Cohen's *d*, Pearson's *r*), indicating how they were calculated |

*Our web collection on statistics for biologists contains articles on many of the points above.*

## Software and code

Policy information about availability of computer code

| Data collection | For repertoire and phylogenetic analyses, genome assemblies were obtained from publically available sources: NCBI (https://www.ncbi.nlm.nih.gov/genome/), Ensembl (https://www.ensembl.org/index.html), and Squalomix (https://transcriptome.riken.jp/squalomi |
|---|---|
| Data analysis | The codes used for the analyses in this paper are available at github.com/obgiorgetti/TCRalpha. |

For manuscripts utilizing custom algorithms or software that are central to the research but not yet described in published literature, software must be made available to editors and reviewers. We strongly encourage code deposition in a community repository (e.g. GitHub). See the Nature Portfolio guidelines for submitting code & software for further information.

## Data

Policy information about availability of data

All manuscripts must include a data availability statement. This statement should provide the following information, where applicable:
- Accession codes, unique identifiers, or web links for publicly available datasets
- A description of any restrictions on data availability
- For clinical datasets or third party data, please ensure that the statement adheres to our policy

The GenBank accession numbers for primary data are PRJNA865512 and PRJNA865921.

# Human research participants

Policy information about studies involving human research participants and Sex and Gender in Research.

**Reporting on sex and gender**

*Use the terms sex (biological attribute) and gender (shaped by social and cultural circumstances) carefully in order to avoid confusing both terms. Indicate if findings apply to only one sex or gender; describe whether sex and gender were considered in study design whether sex and/or gender was determined based on self-reporting or assigned and methods used. Provide in the source data disaggregated sex and gender data where this information has been collected, and consent has been obtained for sharing of individual-level data; provide overall numbers in this Reporting Summary. Please state if this information has not been collected. Report sex- and gender-based analyses where performed, justify reasons for lack of sex- and gender-based analysis.*

**Population characteristics**

*Describe the covariate-relevant population characteristics of the human research participants (e.g. age, genotypic information, past and current diagnosis and treatment categories). If you filled out the behavioural & social sciences study design questions and have nothing to add here, write "See above."*

**Recruitment**

*Describe how participants were recruited. Outline any potential self-selection bias or other biases that may be present and how these are likely to impact results.*

**Ethics oversight**

*Identify the organization(s) that approved the study protocol.*

Note that full information on the approval of the study protocol must also be provided in the manuscript.

# Field-specific reporting

Please select the one below that is the best fit for your research. If you are not sure, read the appropriate sections before making your selection.

☒ Life sciences  ☐ Behavioural & social sciences  ☐ Ecological, evolutionary & environmental sciences

For a reference copy of the document with all sections, see nature.com/documents/nr-reporting-summary-flat.pdf

# Life sciences study design

All studies must disclose on these points even when the disclosure is negative.

**Sample size** | The sample sizes, i.e., the numbers of molecules sequenced for individual Tra and Trb repertoires are given in Supplementary Table 1.

**Data exclusions** | No data were excluded

**Replication** | Where relevant, the biological replicas gave similar results.

**Randomization** | No randomization was done.

**Blinding** | No blinding was done.

# Reporting for specific materials, systems and methods

We require information from authors about some types of materials, experimental systems and methods used in many studies. Here, indicate whether each material, system or method listed is relevant to your study. If you are not sure if a list item applies to your research, read the appropriate section before selecting a response.

## Materials & experimental systems

| n/a | Involved in the study |
|---|---|
| ☐ | ☐ Antibodies |
| ☒ | ☐ Eukaryotic cell lines |
| ☒ | ☐ Palaeontology and archaeology |
| ☐ | ☒ Animals and other organisms |
| ☒ | ☐ Clinical data |
| ☒ | ☐ Dual use research of concern |

## Methods

| n/a | Involved in the study |
|---|---|
| ☒ | ☐ ChIP-seq |
| ☒ | ☐ Flow cytometry |
| ☒ | ☐ MRI-based neuroimaging |

# Antibodies

| Antibodies used | n/a |
|---|---|
| Validation | n/a |

# Animals and other research organisms

Policy information about studies involving animals; ARRIVE guidelines recommended for reporting animal research, and Sex and Gender in Research

| Laboratory animals | Zebrafish (Danio rerio) TU (Tübingen), and TLEK (Tüpfel long fin/Ekkwill) wild-type strains, medaka (Oryzas latipes) and mouse strains are maintained in the animal facility of the Max Planck Institute of Immunobiology and Epigenetics, Freiburg, Germany. |
|---|---|
| Wild animals | Specimens from brownbanded bamboo shark (Chiloscyllium punctatum), gray bichir (Polypterus senegalus), sturgeon (Acipenser ruthenus), West African lungfish (Propterus annectens), and trout (Oncorhynchus mykiss) were obtained from fish dealers. Blood samples from three African bush elephants, Sabie, Tika and Sweni, were obtained from the Wuppertal Zoological Garden. |
| Reporting on sex | Tcr repertoire analyses were carried out without regard to sex. |
| Field-collected samples | see above (Wild animals) |
| Ethics oversight | All animal experiments were performed in accordance with relevant guidelines and regulations, approved by the review committee of the Max Planck Institute of Immunobiology and Epigenetics and the Regierungspräsidium Freiburg, Germany (license AZ 35-9185.81/G-17/79). |

Note that full information on the approval of the study protocol must also be provided in the manuscript.

