## [Peer Review File · Nature]

Manuscript Title: Origin and evolutionary malleability of T cell receptor α diversity

Reviewer Comments & Author Rebuttals

Reviewer Reports on the Initial Version:

Referees' comments:

Referee #1:

There is now considerable evidence to support the notion that a foundational transposition event split the exon of gene encoding a cell surface receptor to provide the evolutionary origin of the V and J segments that are joined to encode antigen receptor chains. Recent work has provided evidence that a primitive transposase indeed has evolved into the RAG enzyme that cleaves at the RSSs of modern V and J segments, allowing them to be joined by the major cellular non-homologous end-joining (NHEJ) pathway to create VJ exons. At some point along the way, D gene segments evolved for some antigen receptor loci to make the process the two-step V(D)J recombination reaction.

In the current study, the authors reasonably base their analysis by considering an existing antigen receptor gene locus that just undergoes one-step V to J joining to be representative of primordial antigen receptor gene ancestors of the more primitive evolutionary pathway. Another critical background point for this study is that during V(D)J recombination in many species, antigen receptor gene diversity is known to be immensely augmented by mechanisms that diversify the critical CDR3 antigen contact region form by V(D)J junctions. Among several such mechanisms is the addition of non-templated N nucleotides by the terminal deoxynucleotidyl transferase (TdT) enzyme. When TdT is developmentally absent (for example, in fetal repertoires in mice or in both fetal and adult repertoires in mice lacking TdT), V(D)J junctional diversity becomes much more limited due to lack of N region diversity and also due to the use of sequence homologies in the RSS-proximal V, D, or J sequences that are used by NHEJ to guide joining.

In this study, the authors focus on examining junctional diversification of Va to Ja junctions during TCRA locus V(D)J recombination in zebrafish versus mice (as well as a few selected other species). Based on studies of junctional diversification of Va to Ja junctions in zebrafish versus mice, the authors explored their basic hypothesis that the presence of, and selection for, a particular sequence composition of the coding regions near the RSS elements of V and J gene segments in evolutionarily early V(D)J recombination pathways resulted from a transposition-generated target-site duplication that guided recurrent NHEJ events. Based on their findings, the authors conclude that the presence and utilization of sequence microhomologies near the termini of Va and Ja gene segments correlates with the variable degrees of junctional diversity of the resulting V-J assemblies in zebrafish and to more variable extent other teleosts versus those of mice. In this regard, they report striking use of germline-encoded microhomology-based in-frame VaJa junctions in zebrafish species versus those of mice. They also demonstrate that zebrafish, and to a more variable extent, other teleosts, have conserved MH-based in-frame sequence-encoding patterns in their Va and Ja segments, whereas those of lobe-finned fish, birds, and mice generally lack such patterns. They conclude that these findings provide an explanation for how somatic diversification became incorporated into, and then maintained in, the immune system of jawed vertebrates. The authors further hypothesize that selection of germline sequence composition of rearranging elements enabled the predictable increase of somatically-generated receptor diversity in step with emerging mechanisms controlling deleterious autoimmunity during evolution.

The basic findings of this study and their suggested implications provide potential new insights into the fascinating subject of how the vertebrate adaptive immune systems evolved and, as such, could be of significant general interest. In particular, the findings of the conserved sequence-repeat based generation of highly restricted TCR α repertoires in zebrafish and other teleosts versus mice is very intriguing. The findings are also consistent with (but do not prove) the authors' speculation that the target-site duplication that accompanied the original transposon insertion flanked both V and J elements and provided a microhomology region that minimized sequence variation at the V to J junction generated by NHEJ. Overall, there are several experimental and conceptual questions that need to be more firmly addressed to allow the findings to be better presented for more in-depth evaluation by the reader and also to clarify the strength of their arguments.

Comments

1. The conclusions of this study, as outlined above, rest on assumptions that V-J recombination in the TCR α locus in zebrafish represents the diversification process of the primordial antigen receptor V to J joining mechanism. Direct V to J recombination also occurs in the TCR γ locus and the Igk and Iglambda light chain loci in mice and apparently in zebrafish as well. The authors do not explain why they chose only to examine the TCR α V to J joining and not TCR γ or IgL chain joining, which according to their definition, could also potentially be primitive in zebrafish. At a minimum, it is essential that the authors discuss why, in the context of the rationale for considering the zebrafish TCR α locus to be primordial as a founding principle of the study, they did not apply this rationale to other V to J rearranging zebrafish loci at the outset of the work. While it may take additional experimental effort to address the repertoires generated from these loci in zebrafish, it should be straightforward to examine relevant sequence signatures in the corresponding V and J loci in zebrafish versus mice.

2. The authors raise the interesting speculation that MH-mediated recombination generates recurrent antigen receptors that may function more like pattern recognition receptors and which are not autoreactive. In this context, they also suggest that as receptor diversification increased, a tolerance control mechanism evolved in step to control potentially deleterious effects of self-reactive antigen receptors resulting that are a by-product of increased diversification. These notions appear to be presented as major findings of the study as indicated in the concluding line of the abstract and by featuring them as the major topic of the long (for Nature) discussion. Without experimental data to support such conclusions, these interesting speculations should probably be condensed to a few lines in the discussion. Also, the abstract is very vague with respect to describing the main experiments presented in the paper. The authors should consider rewriting much of the text to make it more specific and logical.

3. In Fig. 1, the authors compared the TCR α repertoires in zebrafish and mouse. But the experiments have several potential issues that the authors should address. First, the repertoire data for zebrafish were derived from sequencing of whole fish total RNA, which includes contributions from both developing thymocytes and mature T cells. However, the repertoire data for mouse were derived from analysis of total RNA from splenocytes, which include only mature T cells. Thus, the data may not be directly comparable. A perhaps more significant issue is that the in-frame and out-of-frame repertoires were generated by RNA-seq. It is known from the literature that RNA sequences from out-of-frame V(D)J recombination events can be greatly under-represented in RNA-seq analyses due to nonsense-mediated decay. To circumvent this potential limitation, high-throughput genomic DNA sequencing has been used to obtain accurate representation of in-frame and out-of-frame rearrangements in many studies. Have the authors demonstrated that out-of-frame zebrafish RNAs do not result in nonsense-mediated decay and that this could at least partially contribute to data in Fig. 1b? Also, why does the non-selectable repertoire in mouse include CDR3 length of in-frame joints? Do these joints contain stop codons? In this regard, in Fig. 1b the authors should describe the method used to generate the non-selectable repertoire in the mouse splenic B cell data, as this is not explained anywhere in the

paper. Also, depending on how this data was generated, could there not also have been an influence of NSMD? In this regard, it is hard to find a comparison of the relative sequence recovery in the mouse experiments (assuming non-productive rearrangements also involved an in-frame stop codon allele in a Ca exon) with that of a normal productive allele.

4. The authors should clarify the sequence data in Fig. 1c. Are they from the selectable repertoire, non-selectable repertoire, in-frame sequence, or out-of-frame sequence?

5. The meaning of the plot in Fig. 1e needs clarification. It is not clear what is presented in this figure and how exactly to interpret the plots. The figure legend or presentation in the text is not helpful to this reader.

6. The authors state in various places in the manuscript that "TdT blocks microhomology-dependent recombination". However, this is likely not generally the case. What is the evidence that TdT "blocks" microhomology-mediated end-joining *in vivo* during V(D)J recombination or NHEJ more generally? It has been considered likely that TdT-added N regions can provide homologies with N regions, P elements, or with coding sequences on the other participating ends that are undetectable because it is not possible to know what nucleotides were added by TdT. Such N-region-provided microhomologies might even lead to some apparently "blunt" end joins. It would seem more appropriate to indicate that TdT impedes the utilization of microhomologies at the coding ends of two gene segments being joined (perhaps by generating new microhomologies or extending the junction sequences beyond the germline encoded microhomologies, etc.). While it is not necessarily a major point, for accuracy, it is suggested that the authors should incorporate this point into the text and figures.

7. In Fig. 3c, the authors show that the CDR3 diversity is correlated with the fraction of TCRa CDR3 sequences without non-templated nucleotide addition. But if the authors wish to clarify the impact of sequence signature on CDR3 diversity, the x-axis should be "conservation of sequence signature".

8. In Fig. 3a, the authors plotted all V and J sequences together. But the mouse V and J sequences are more variable than those in other species. It may be better to classify the mouse V and J sequences into several subgroups based on the sequence similarity and then compare the subgroups to other species.

9. Fig. 3e shows that in zebrafish, TCRa joins lack N regions whereas TCRb joins contain them. Also, Ext. Data Fig. 9 shows that some of species of fish that have the conserved sequence signatures also have a large portion of CDR3s with non-templated nucleotides. The authors should clarify more precisely whether they are assuming TdT (or a related DNA polymerase) is developmentally regulated during the rearrangement of different TCR loci and whether TdT is differentially regulated during TCRa rearrangement in different zebrafish species. With respect to the former possibility, a potentially analogous situation in mammals involves TdT expression in human but not mouse pre-B cells. Consequently, the IgL repertoire in humans is much more complex than that of mice which frequently use microhomologies to mediate recurrent VL to JL joins. It would be useful to assay TdT expression during T cell development in zebrafish to strengthen their arguments. It may also be worth mentioning the evolutionarily conserved organization of the TCRd locus within the TCRa locus from zebrafish to mice and if possible better compare rearrangement properties of the TCRd locus (which already evolved D segments) to that of its host TCRa locus in zebrafish.

10. Some words and phrases are vague throughout the manuscript. Some specific examples: In the discussion, the sentence in lines 281-282 is not correct. Thus, TdT expression in fact increases overall recombination diversity and does not "diminish overall recombination diversity". Also, as mentioned, TdT expression suppresses "recurrent or dominant" recombination that involves microhomologies in the RSS-proximal regions of coding ends being joined, but does not

necessarily block NHEJ using microhomologies per se. Other examples are in line 269, in which it is not clear what exactly 'epistasis' refers to as used, and in line 288, the phrase 'In this situation is unclear (which situation?)

Referee #2 (Remarks to the Author):

The authors propose that the ancestral RAG recombination process occurs via microhomologous recombination. This would have avoided the risk of harmful self-recognition that would have occurred without a mechanism of negative selection (via the thymus for example) that was probably lacking in the vertebrate ancestor. The region of microhomology corresponds to the TSD generated by the transposon RAG, which has been domesticated in jawed vertebrates as the RAG gene.

The analysis presented by the authors on the TCR alpha repertoire across jawed vertebrates is consistent with the hypothesis. They show that recombination of the RAG process occurs via microhomologous recombination in cartilaginous vertebrates, bony fishes but not in tetrapods. As cartilaginous vertebrates and paraphyletic bony fishes (along with tetrapods) conclude that the ancestral state of recombination occurs via microhomologous recombination.

1) Could the authors better explain why they think recombination via microhomologous regions is ancestral? I'm not sure non-specialists would understand.

2) The authors have written in several places that microhomologous recombination is due in part to the inefficiency of TDT (Terminal Deoxyribonucleotidyl Transferase). Microhomologous recombination occurs in the case of the zebrafish T-cell receptor alpha. As far as I understand, this is not the case for the zebrafish T-cell receptor beta (please clarify this point). If so, is it possible that TDT is not expressed between the DN4 stage and the double-positive stage (where the Valpha Jalpha recombination occurs) and is expressed in the stage where the Vbeta Jgamma recombination occurs? If TDT is not expressed between the DN4 and double positive stages (where Valpha Jalpha recombination occurs) how do the authors explain the "rogue" elements Va16-1 and Ja5?

3) Page 141: Several transposons closer to RAG than to Transib have been described. The TSD is the same size as in the case of Transib (PMID: 27293192).

4) Even if the TDT is absent after the excision of the intermediate DNA, hairpins are formed in the surrounding DNA. Next, the hairpins are treated with Artemis endonuclease, effecting an asymmetric opening of the hairpin and leading to a palindromic variation of P nucleotides at the junction. This is described (of course) not only in the case of RAG but also in the case of the transposon RAG (PMID: 27293192). Do the authors see this (P nucleotide diversity) in their analysis on the TCR alpha repertoire (even if this is found in few cases)?

Minor comments

5) L255: What do the authors mean by "The radical deviation with Darwinian selection" (it seems to me that they mean substitutions versus recombination)? Please clarify.

6) L274: In the case of iNKT and MAIT cell receptors, does V(D)J recombination occur via the microhomologous regions?

7) Recombination via the regions of microhomologies, after transposon excision creating hairpins, has been described (see for example PMID: PMC321453). It could be the same for the RAG

transposon. This should be discussed in the context of the authors' results.

Referee #3:

The article submitted by Giorgetti, O'Meara, Schorpp, and Boehm provides a new understanding of the evolution of adaptive immunity. They make the novel observation (as far as I know), that most TCR-alpha VJ rearrangements in zebrafish arise via microhomology-directed, non-homologous end joining, and thus, even in the absence of selection, alpha-chain VJ junctions are largely in-frame. In addition, there are almost no non-templated sequences in the mature repertoire of zebrafish. Both of these observations are consistent with a lack of TdT expressed during the process of zebrafish alpha-chain rearrangements. This means that the TCR a-chain in zebrafish is limited in diversity, in stark contrast to the parallel analysis carried out on mice. Since there is a TdT ortholog expressed in the thymic cortex of zebrafish, could the authors comment on the lack of an apparent TdT influence?

The authors show this by analyzing zebrafish that have a heterozygous mutation in the alpha locus such that one allele is functional and the other undergoes recombination with no possible cellular selection. They refer to this as the non-selectable repertoire, and show that it is almost entirely in-frame. When they displaced the position of the RSS elements, this in-frame pattern disappeared showing that the position of a micro-homology is essential to the conservation pattern.

The analysis goes to show that these observations are true for three other teleost species, and in a wider analysis, the conserved microhomologies are present in cartilaginous, basal, and teleost fishes, whereas it is missing lungfish and mammals. The authors propose that a "TG" motif is part of a primordial sequence pattern that directs alpha-chain gene assembly in cartilaginous and basal fishes, and teleosts.

All of these very extensive analyses are very well documented in extensive supplementary material and clearly presented in clean (and sometimes colorful) expository writing.

Based on these findings, they present a theoretical answer to a conundrum of the adaptive/acquired immune system that is, how would a highly diverse repertoire of lymphocyte receptors coincidentally evolve with mechanisms to ensure self-tolerance? Their novel hypothesis is that the adaptive immune system in the proto-vertebrate (both gnathostomes and agnatha) was restricted in sequence diversity, and presumably most of the resulting receptors were not specific for self-molecules present in the species. With time, cellular tolerance mechanisms provided a selective advantage, even to the protovertebrates, that eventually allowed sequence diversity to evolve to that is found in birds, skates and mammals.

A prediction, based on the authors' theory of limited diversity and self-tolerance, is that in the protovertebrate, both alpha and beta sequences would be restricted since of course the beta chains can contribute equally to antigen peptide specificity. For some reason this limitation in alpha chain diversity has been maintained in zebrafish for the alpha chain, but it appears to have evolved in the b-chain locus to look like the diversity found in mammals (Fig. 1e; Supplementary Tables 1,3). At least that was my initial take; however, in figure 3E, the ordinate is "mean microhomologies per molecule", whereas the text describes non-templated nucleotide insertions". Could the authors please clarify this for me and indicate whether the zebrafish b-chain locus shows evidence of restricted diversity and a lack of TdT (and again the presence of thymic TdT)? Why have the teleosts retained the protovertebrate VJ recombination and repair mechanism? For example, do zebrafish have stunted mechanisms of tolerance (if this is known)?

Another issue is B cell receptors and T cell receptors. Do the antibody loci in teleosts also show restricted diversity, or have these antibody-encoding loci evolved to show high levels of diversity and alongside effective tolerance mechanisms?

In the Discussion, there is a statement, "Although TdT has been shown to suppress microhomology-guided recombination and diminish overall recombination diversity^{29,32-34}, which in fetal life leads to a more stereotypical repertoire..." What is meant by diminish the overall recombination diversity? I would think that TdT increases the overall receptor diversity? Can the authors please clarify.

Overall, this is a spectacular set of findings and accompanying analyses. I think the theory is novel and interesting, and perhaps testable. I highly recommend publication in Nature.

Referee #4:

The manuscript by Giorgetti et al. is an intriguing exploration of the hypothesis that tiny stretches of DNA sequence homology near the ends of recombining segments used in V(D)J recombination were key in allowing the emergence of such a stochastic system of receptor generation without overly detrimental self-recognition. Here they present a robust set of comparative genomic analyses of the T cell receptor alpha loci of diverse species that supports the hypothesis. Additionally, they find that TCR alpha repertoire diversity as expressed at the mRNA level correlates with the extent of this microhomology.

The authors test this hypothesis with a great deal of work, including CRISPR-engineered zebrafish, a remarkably wide comparative approach involving sampling everything from sharks to elephants, repertoire sequence analysis in species that must have required significant troubleshooting of amplification parameters, and a Herculean amount of genomic and amplicon bioinformatics. Even with the IMGT database and the most cutting-edge tools, those of us who look at antigen receptor loci for a living know that assemblies in these regions are wrought with pitfalls... there is no way around much manual analysis, checking and annotation.

Twenty years after Tonegawa demonstrated somatic gene rearrangement in lymphocyte antigen receptors, Sam Schluter and Jack Marchalonis described the "Big Bang" that appeared to be the genesis of our combinatorial adaptive immune system (Schluter et al., *Dev. Comp. Immunol.*, 1999) in sharks. Nearly twenty years later David Schatz and team definitively showed us how the horizontal transposon event evolved into the RAG recombinase structurally (Liu et al., *Nature*, 2019). Now, for the first time, that Big Bang seems clear... and a bit more of a rolling start?

The steps are now much more clear how on top of the historically-gazing, reactive, innate immune system a radically preemptive, forward-gazing, anticipatory system evolved.

This is likely not an easy read, even for those somewhat versed in the immunogenetics of lymphocyte antigen receptors and theories on their origins. However, ample background is not possible in the format. The following items I think bear consideration before this manuscript is ready for the readership of Nature.

Major points

The more impactful the work the longer the list of obvious new questions and experiments mandated, but I know a line has to be drawn somewhere. These first two major questions many readers will have spring from the focus on the alpha chain, with little explanation why. While I do not think deeper exploration of other antigen receptor loci is necessary for this paper, I do hope a little discussion and potential next hypotheses can be incorporated into the tight word limits. It will be important to help the general reader see the broader impact of this work.

1) In the first paragraph of the introduction the authors acknowledge that the hypothetical gene targeted by the ancestral (transib) transposon is considered the founding member of the AgR of

jawed vertebrates, yet alpha is studied instead of gamma (or even immunoglobulin light chain lambda), which is sometimes suggested to be the ancestral chain of our lymphocyte antigen receptors. Were loci encoding other chains than TCR alpha and beta explored? The work on TCR alpha is deep and complete (my second paragraph above), which is why I wonder why the jump to this locus instead of TCR gamma which would be simpler in many ways (fewer V and J elements, no confounding delta use (line 232)). It is recognized that the D segment using loci (immunoglobulin heavy chain, TCRS beta and delta) are not the most parsimonious candidates for the ancestral.

2) Related to point #1 above... This evolution of the TCR alpha V and J arrays presumably was in lock step with the MHC allelic "repertoires" in these early vertebrates. To my knowledge, restriction of canonical alpha/beta T cell receptor to MHC peptide presentation is extremely early in the system... MHC is both polygenic and polymorphic in all extant cartilaginous jawed fishes, and even lamprey VLRs may have a receptor and lymphocyte subset transcriptionally orthologous to gnathostome alpha beta T cells. Do the authors see the CDRs 1 and 2 of the TCR alpha variable segments quickly evolving for recognition of the alpha helices of the MHC peptide binding domains and the junctional diversity of CDR3 for the peptide?

3) Perhaps too late now, but the title does not quite capture the gestalt of the paper. Isn't it suggesting the origin of tolerance in our adaptive immune system... and actually challenging the last of Sir McFarlane Burnet's four postulates of clonal selection? Perhaps autoreactive lymphocytes do not have to be negatively selected if they do not arise due to restricted junctional diversity.

4) Is it possible that such a mechanism was not necessary? Seems the antigen universe of non-self/potential pathogen was likely larger than the self Ag repertoire to begin with, tolerance mechanisms would just be value added. Lines 47-50 may not be right, although the data here suggest it was advantageous for the development of our adaptive system half a billion years ago.

Minor points

- Are the four patterns of microhomology passed through speciation events and higher taxa? (Do the same V-J pairs with the same microhomologies exist between two teleosts, or between a shark and a lungfish?) I am questioning how species-specific the public repertoires actually are (line 173).

- AID-mediated somatic hypermutation has been described at nurse shark TCR alpha, occurring in the thymus (Ott et al., ELife, 2018). Does this present a problem for the junctional repertoire restriction mechanism described here? The TCR SHM could have evolved much later, at least one way to reconcile.

- I am intrigued by the range of the amphibian RSS J frame conservation (line 241) and how it does not quite fit in with the trend (Fig. 4b). Any ideas as to why? Louis du Pasquier and others have studied the *Xenopus* immune system pre- and post-metamorphosis, a radical reboot. Perhaps it is related to that? And does it track with size (limited T cell compartment size in very small tadpoles of some species may limit?). I am curious if there is a relation across the range of species between body size and RSS conservation.

- Line 235, what is "c.f."?

- Line 289, the first "are" needs to be removed.

- Line 423, looks like an aberrant bold in "length".

- Line 570, last word should be "the" not "then".

- Why were ten species excluded from Ext. Data Fig. 12?

- Fig. 3a, would be useful to align the amino acid motif under or above the nucleotide seqs for orientation (C of the V, how far 5'/amino-terminal of the FGxG of the J).

Author Rebuttals to Initial Comments:

Response to reviewers' comments.

We thank the reviewers for their careful assessment of our work, and the time they have invested in providing critical feedback and constructive critiques. This greatly helped us in improving the clarity of exposition and the presentation of our results in the revised version of the manuscript. We respond to each of the points below.

Referees' comments:

Referee #1:

There is now considerable evidence to support the notion that a foundational transposition event split the exon of gene encoding a cell surface receptor to provide the evolutionary origin of the V and J segments that are joined to encode antigen receptor chains. Recent work has provided evidence that a primitive transposase indeed has evolved into the RAG enzyme that cleaves at the RSSs of modern V and J segments, allowing them to be joined by the major cellular non-homologous end-joining (NHEJ) pathway to create VJ exons. At some point along the way, D gene segments evolved for some antigen receptor loci to make the process the two-step V(D)J recombination reaction.

In the current study, the authors reasonably base their analysis by considering an existing antigen receptor gene locus that just undergoes one-step V to J joining to be representative of primordial antigen receptor gene ancestors of the more primitive evolutionary pathway. Another critical background point for this study is that during V(D)J recombination in many species, antigen receptor gene diversity is known to be immensely augmented by mechanisms that diversify the critical CDR3 antigen contact region form by V(D)J junctions. Among several such mechanisms is the addition of non-templated N nucleotides by the terminal deoxynucleotidyl transferase (TdT) enzyme. When TdT is developmentally absent (for example, in fetal repertoires in mice or in both fetal and adult repertoires in mice lacking TdT), V(D)J junctional diversity becomes much more limited due to lack of N region diversity and also due to the use of sequence homologies in the RSS-proximal V, D, or J sequences that are used by NHEJ to guide joining.

In this study, the authors focus on examining junctional diversification of Va to Ja junctions during TCRA locus V(D)J recombination in zebrafish versus mice (as well as a few selected other species). Based on studies of junctional diversification of Va to Ja junctions in zebrafish versus mice, the authors explored their basic hypothesis that the presence of, and selection for, a particular sequence composition of the coding regions near the RSS elements of V and J gene segments in evolutionarily early V(D)J recombination pathways resulted from a transposition-generated target-site duplication that guided recurrent NHEJ events. Based on their findings, the authors conclude that the presence and utilization of sequence microhomologies near the termini of Va and Ja gene segments correlates with the variable degrees of junctional diversity of the resulting V-J

assemblies in zebrafish and to more variable extent other teleosts versus those of mice. In this regard, they report striking use of germline-encoded microhomology-based in-frame VaJa junctions in zebrafish species versus those of mice. They also demonstrate that zebrafish, and to a more variable extent, other teleosts, have conserved MH-based in-frame sequence-encoding patterns in their Va and Ja segments, whereas those of lobe-finned fish, birds, and mice generally lack such patterns. They conclude that these findings provide an explanation for how somatic diversification became incorporated into, and then maintained in, the immune system of jawed vertebrates. The authors further hypothesize that selection of germline sequence composition of rearranging elements enabled the predictable increase of somatically-generated receptor diversity in step with emerging mechanisms controlling deleterious autoimmunity during evolution.

The basic findings of this study and their suggested implications provide potential new insights into the fascinating subject of how the vertebrate adaptive immune systems evolved and, as such, could be of significant general interest. In particular, the findings of the conserved sequence-repeat based generation of highly restricted TCR α repertoires in zebrafish and other teleosts versus mice is very intriguing. The findings are also consistent with (but do not prove) the authors' speculation that the target-site duplication that accompanied the original transposon insertion flanked both V and J elements and provided a microhomology region that minimized sequence variation at the V to J junction generated by NHEJ. Overall, there are several experimental and conceptual questions that need to be more firmly addressed to allow the findings to be better presented for more in-depth evaluation by the reader and also to clarify the strength of their arguments.

Response: We thank the reviewer for her/his careful and positive assessment of our study and its implications.

Comments

1. The conclusions of this study, as outlined above, rest on assumptions that V-J recombination in the TCR α locus in zebrafish represents the diversification process of the primordial antigen receptor V to J joining mechanism. Direct V to J recombination also occurs in the TCR γ locus and the Igk and Iglambda light chain loci in mice and apparently in zebrafish as well. The authors do not explain why they chose only to examine the TCR α V to J joining and not TCR γ or IgL chain joining, which according to their definition, could also potentially be primitive in zebrafish. At a minimum, it is essential that the authors discuss why, in the context of the rationale for considering the zebrafish TCR α locus to be primordial as a founding principle of the study, they did not apply this rationale to other V to J rearranging zebrafish loci at the outset of the work. While it may take additional experimental effort to address the repertoires generated from these loci in zebrafish, it should be straightforward to examine relevant sequence signatures in the corresponding V and J loci in zebrafish versus mice.

Response: The reviewer raises an interesting point and is correct to observe that TCR α , TCR γ , and the Ig light chain gene loci all retain the presumed primordial V-J structure. We focused on TCR α for several reasons:

(1) The *TCRg* locus often consists of only very few *V* and *J* elements. For instance, in the mouse, 7 *Vg* and 4 *Jg* elements are recognized; in the zebrafish, 7 *Vg* and 7 *Jg* are known. By contrast, the *TCRa* locus usually comprises dozens of *V* and *J* elements (mouse: 64 *Va* and 45 *Ja* elements respectively; zebrafish: 124 *Va* and 134 *Ja* elements respectively). Based on the data deposited in the IMGT database (<https://www.imgt.org/>), differences in the numbers of *V* and *J* segments between *TCRg* and *TCRa* also hold for 11 other species, as depicted in **new Supplementary Figure 1a**.

(2) It has now been shown for many species (such as squamates [Morissey et al., J. Immunol. 208, 1960, 2022]) that they have lost the genes for the TCRgd receptor. Apart from introducing bias in broad phylogenetic surveys, the statistical robustness of any conclusion about conserved sequence elements is inversely proportional to the numbers *V* and *J* elements associated with *TCRg*. Hence, we consider the inferences derived from the analyses of the *TCRa* loci to be less prone to random fluctuations and therefore more meaningful.

With respect to the analysis of *IgL* genes, two reasons suggested to us that they may not be the best object of study at this point in time:

(1) *IgL* loci commonly present with few *J* elements, although the number of *V* elements varies considerably (**new Supplementary Fig. 1b**). By way of example, a total of 51 *V_L* genes and 17 *J_L* (all 5 loci combined) were described for zebrafish (Zimmerman et al., Dev. Comp. Immunol. 32, 421, 2008), and 146 *V_L* and 12 *J_L* for mouse (kappa and lambda loci combined; IMGT database).

(2) Since the number and structure of *IgL* gene loci varies among vertebrate species, it is not straightforward to establish orthology, complicating inferences based on phylogenetic relationships.

As alluded to by the reviewer, the emergence of *D* elements is likely a secondary step in the evolution of *Tcr* (and of course also the *Igh*) genes, indicating that the V-J configuration is closest to the situation of the primordial V-J antigen receptor gene.

In view of the above summary, we considered the *TCRa* locus to be the best target for our investigation, as it is a constant companion of canonical adaptive immune systems in jawed vertebrates (with the exception of a small number of species of deep-sea anglerfishes that we recently found to have lost canonical adaptive immunity after pseudogenization of RAG genes [Swann et al., Science 369, 1608, 2022]).

Moreover, since one of the main motivations of our study was to enable a future exploration of the connection between antigen receptor diversity and tolerance mechanisms, we felt it would be best to focus on a system that rests on a well-studied principle for quality control; i.e., MHC peptide presentation. We have summarized the considerations guiding the choice of *Tcr* as the focus of the present study in **new Supplementary text**.

In response to the reviewer's concern, we have compiled and added additional information on the structure of *V* and *J* elements of *TCRg* and *IgL* loci in zebrafish and mouse, with respect to the potential presence of conserved sequence patterns at the ends of *Vg* and *Jg* elements. From this analysis (presented in **new Supplementary Figures 2-5**), we conclude the following:

(i) The positions of the RSS relative to the conserved cysteine codon in *Vg* elements of zebrafish vary (**new Supplementary Fig. 2**), in stark contrast to the situation of *Va* elements in this species (**Extended Data Fig. 1**). In contrast to the situation of *Ja* of zebrafish, the *Jg* elements lack a consistent sequence motif downstream of the RSS elements (**new Supplementary Fig. 2**). This leads to a much greater length variation of the CDR3 regions of *TCRg* assemblies (**new Supplementary Fig. 2**), in contrast to the stereotyped 3-nucleotide step pattern observed in *tcra* assemblies (**Extended Data Figure 3**). The sequence analysis of the *TCRg* assemblies does however suggest the possibility of occasional microhomology-directed recombination, although this phenomenon is much less pronounced than in the *TCRa* assemblies, at least based on the few zebrafish *tcrg* assemblies that have been reported (**new Supplementary Fig. 2**).

(ii) For the mouse *TCRg* elements, the structure of the germ-line *Vg* and *Jg* elements also suggests the possibility of microhomology-directed recombination (**new Supplementary Fig. 3**), which has already been experimentally demonstrated (Zhang et al., *Immunity* 3, 439, 1993), a finding that we had referred to in the original version of the manuscript.

(iii) With respect to the situation of *IgL* loci in zebrafish, we provide sequence alignments of *V_L* and *J_L* elements for the five genomic clusters described by Zimmerman et al. (*Dev. Comp. Immunol.* 32, 421, 2008) (**new Supplementary Fig. 4**). This analysis shows the presence of similar sequence motifs at the ends of *V* and *J* elements for some but not all clusters. For the mouse (**new Supplementary Fig. 5**), it appears that matching sequence motifs are detectable as well.

In conclusion, the analysis of additional V-J type *TCR* and *IG* genes indicates that microhomology-directed recombination appears to be a general feature of V-J type antigen receptor loci, although locus-specific and species-specific differences are recognizable.

Overall, this extended survey is in line with our hypothesis that the original V-J gene exhibited this feature, and that it became degraded to variable extends over the course of evolution, and in response to the divergent functional requirements imposed on the different antigen receptor genes.

We thank the reviewer for giving us the opportunity to explain more explicitly the basis for our study.

2. The authors raise the interesting speculation that MH-mediated recombination generates recurrent antigen receptors that may function more like pattern recognition receptors and which are not autoreactive. In this context, they also suggest that as receptor diversification increased, a tolerance control mechanism evolved in step to control potentially deleterious effects of self-reactive antigen receptors resulting that are a by-product of increased diversification. These notions appear to be presented as major findings of the study as indicated in the concluding line of the abstract and by featuring them as the major topic of the long (for Nature) discussion. Without experimental data to support such conclusions, these interesting speculations should probably be condensed to a few lines in the discussion. Also, the abstract is very vague with respect to describing the main experiments presented in the paper. The authors should consider rewriting much of the text it to make it more specific and logical.

Response: We thank the reviewer for this comment. Our intention of elaborating on the relationship between AgR diversity and requirement of tolerance induction was to put the

motivation and outcome of our study into context. However, we agree that it would be best to remove this aspect from the abstract and to leave it to the discussion section. We have done so in the revised manuscript.

3. In Fig. 1, the authors compared the TCR α repertoires in zebrafish and mouse. But the experiments have several potential issues that the authors should address. First, the repertoire data for zebrafish were derived from sequencing of whole fish total RNA, which includes contributions from both developing thymocytes and mature T cells. However, the repertoire data for mouse were derived from analysis of total RNA from splenocytes, which include only mature T cells. Thus, the data may not be directly comparable.

Response: We thank the reviewer for raising this issue. In the zebrafish experiment, we have generated cDNAs from whole body mRNA, which thus includes T cells from thymus and peripheral tissues. The fish used for this experiment are heterozygous for a mutant *tcra* allele. It was generated by CRISPR/Cas9-mediated mutagenesis to induce a small deletion causing a frame-shift/premature stop coding in the first exon of the constant region gene. This constellation allows us to look at sequences from both the wild-type and mutant alleles in the same individual. It is important to note that rearrangements on the allele with the disrupted constant region cannot be subjected to post-assembly selection, because no protein can be made. Therefore, rearrangements on this allele are representative of the pre-selection repertoire (regardless of the source of T cell populations). We can thus directly compare the structure of the CDR3 regions of such non-selectable sequences with their selectable cousins that possess a wild-type constant region. To explain the difference between selectable and non-selectable alleles that are studied in Fig. 1a,b more clearly, we have added relevant schematics. The striking observation was that for zebrafish, but not for mouse, the pattern of CDR3 structures was identical between selectable and non-selectable alleles, indicating a dominant influence of the rearrangement process on the structure of resulting *tcra* sequences.

A perhaps more significant issue is that the in-frame and out-of-frame repertoires were generated by RNA-seq. It is known from the literature that RNA sequences from out-of-frame V(D)J recombination events can be greatly under-represented in RNA-seq analyses due nonsense-mediated decay. To circumvent this potential limitation, high-throughput genomic DNA sequencing has been used to obtain accurate representation of in-frame and out-of-frame rearrangements in many studies. Have the authors demonstrated that out-of-frame zebrafish RNAs do not result in nonsense-mediated decay and that this could at least partially contribute to data in Fig. 1b?

Response: The reviewer raises a valid point. However, we know from our analysis that non-sense mediated decay (NMD) of RNAs does not interfere with the analysis of V-J assemblies in the **zebrafish** mutants. When considering the number of UMIs as a representative of the number of mRNA molecules, we found that for heterozygous fish, ~48% of molecules in the repertoire originated from the wild-type allele, and ~52% from the mutant allele. The comparison between selectable and non-selectable repertoire of the **mouse** also indicates that NMD does not have a great impact on the representation of out-of-frame *Tcra* assemblies; otherwise, the green bars (that is, sequences with an in-frame

CDR3 region, but non-productive because of the constant region mutation) in the right panel of Fig. 1b would not add up to 35.6% of all sequences, close to the expected 1/3 value for random outcomes. In retrospect, we realize that we should have mentioned this aspect in the methods section, which we have done now in the revised version.

Also, why does the non-selectable repertoire in mouse include CDR3 length of in-frame joints? Do these joints contain stop codons? In this regard, in Fig. 1b the authors should describe the method used to generate the non-selectable repertoire in the mouse splenic B cell data, as this is not explained anywhere in the paper. Also, depending on how this data was generated, could there not also have been an influence of NSMD? In this regard, it is hard to find a comparison of the relative sequence recovery in the mouse experiments (assuming non-productive rearrangements also involved an in-frame stop codon allele in a Ca exon) with that of a normal productive allele.

Response: We thank the reviewer for alerting us to the omission of details concerning the repertoire analysis of mouse *TCRa*. We have amended the text in the revised version to make explicit that the mouse *TCRa* is rendered non-functional by insertion of a *neo* gene cassette into the first exon of the constant region gene (Mombaerts et al., Nature 360, 225, 1992). The *Tcra* repertoire of mutant mice was generated from thymus cDNA of homozygous null mice; the *Tcra* repertoire of wildtype mice was generated from splenocytes. Unfortunately, this information was not given in the methods section; we have now corrected this oversight. With respect to the question about NMD, we refer the reviewer to our response to point#3 above.

4. The authors should clarify the sequence data in Fig. 1c. Are they from the selectable repertoire, non-selectable repertoire, in-frame sequence, or out-of-frame sequence?

Response: We thank the reviewer for requesting this clarification. We now state in the figure legend that the sequences are derived from wild-type animals; therefore, out-of-frame sequences make up a minor fraction of the whole repertoire (see Fig. 1a).

5. The meaning of the plot in Fig. 1e needs clarification. It is not clear what is presented in this figure and how exactly to interpret the plots. The figure legend or presentation in the text is not helpful to this reader.

Response: We apologize to the reviewer if our description of this panel was not clear. We have changed the label on the x-axis to "Publicity", to indicate the number of individual fish in which a particular clonotype is found. In this way, the drastic difference in the degree of publicity of the *tcrA* and *tcrB* repertoires becomes obvious. Because the outcome of *tcrA* rearrangements are to a large degree predictable, it is no surprise, but needed to be shown, that different individuals share a large fraction of their clonotypes. Therefore, we have also amended the description of the y-axis accordingly; it now reads: 'Fraction of the repertoire (clonotype usage weighted by UMI)'. We have rephrased the text describing this figure panel accordingly.

6. The authors state in various places in the manuscript that "TdT blocks microhomology-dependent recombination". However, this is likely not generally the case. What is the evidence that TdT "blocks" microhomology-mediated end-joining in vivo during V(D)J recombination or NHEJ more generally? It has been considered likely that TdT-

added N regions can provide homologies with N regions, P elements, or with coding sequences on the other participating ends that are undetectable because it is not possible to know what nucleotides were added by TdT. Such N-region-provided microhomologies might even lead to some apparently “blunt” end joins. It would seem more appropriate to indicate that TdT impedes the utilization of microhomologies at the coding ends of two gene segments being joined (perhaps by generating new microhomologies or extending the junction sequences beyond the germline encoded microhomologies, etc.). While it is not necessarily a major point, for accuracy, it is suggested that the authors should incorporate this point into the text and figures.

Response: We thank the reviewer for this comment. We agree that our statement may have been too categorical; accordingly, we have toned down/rephrased the text to reflect the fact that TdT impedes, but does not block the usage of microhomologies during formation of the coding joints.

7. In Fig. 3c, the authors show that the CDR3 diversity is correlated with the fraction of TCRa CDR3 sequences without non-templated nucleotide addition. But if the authors wish to clarify the impact of sequence signature on CDR3 diversity, the x-axis should be “conservation of sequence signature”.

Response: Thank you for alerting us to this issue. In Fig. 2, we described the four homology patterns in zebrafish, all resulting in CDR3 sequences without insertions. The purpose of Fig. 3c was to highlight the fact that the fraction of assemblies without insertions varies in phylogeny. As the reviewer correctly noticed, much (but not all) of this can be explained by a “conservation of sequence signature”. The impact of conserved sequence patterns can be gleaned from **new panels e and f** added to **Extended Data Figure 10**, which address the concern of the reviewer. In these new figure panels, we plot the fractions of V and J elements in 10 species that conform to the characteristic TGNNGCC and TGA(C/A)T patterns respectively, to complement the presentation in Fig. 3a. This is another way to visualize the degradation of microhomology-supporting sequences in the vicinity of the RSS in lobe-finned fishes and mammals.

8. In Fig. 3a, the authors plotted all V and J sequences together. But the mouse V and J sequences are more variable than those in other species. It may be better to classify the mouse V and J sequences into several subgroups based on the sequence similarity and then compare the subgroups to other species.

Response: We thank the reviewer for this suggestion and have now provide the relevant data, which we have added to **Extended Data Figure 10** as **new panels e and f**. From this figure, one can directly deduce that, for instance, about 20% of mouse *Ja* elements conform to the 'primordial' pattern, which is indeed hidden among the more variable composition of the majority of *Va* elements.

9. Fig. 3e shows that in zebrafish, TCRa joins lack N regions whereas TCRb joins contain them. Also, Ext. Data Fig. 9 shows that some of species of fish that have the conserved sequence signatures also have a large portion of CDR3s with non-templated nucleotides. The authors should clarify more precisely whether they are assuming TdT (or a related DNA polymerase) is developmentally regulated during the rearrangement of different TCR loci and whether TdT is differentially regulated during TCRa rearrangement in

different zebrafish species. With respect to the former possibility, a potentially analogous situation in mammals involves TdT expression in human but not mouse pre-B cells. Consequently, the IgL repertoire in humans is much more complex than that of mice which frequently use microhomologies to mediate recurrent VL to JL joins. It would be useful to assay TdT expression during T cell development in zebrafish to strengthen their arguments.

Response: We appreciate the concern of the reviewer with regards to the expression pattern of TdT in zebrafish and other teleosts. We agree that the most likely explanation for the difference between *tcra* and *tcrb* assemblies is the presence of absence of TdT (or perhaps DNA polymerase mu) during the assembly process. Unfortunately, the developmental progression of T cells (and B cells for that matter) in zebrafish is not well defined. Therefore, it is currently impossible to isolate teleost thymocytes that would mirror the successive stages of mouse T (or B) cell development. Hence, although important, the analysis of this question is not trivial, and must await the development of stage-specific isolation procedures which are beyond the scope of our manuscript. Nonetheless, we note that others have demonstrated differential expression of TdT by in situ hybridization of thymus sections (Beetz et al., Immunogenetics 59, 735, 2007) and more recently by scRNA-seq in T cells (and B cells) (Rubin et al., J. Exp. Med. 219, e20220038, 2022), suggesting that the developmental regulation of TdT may be a conserved, yet evolutionarily malleable, feature of lymphocyte development. We have introduced reference to this work in the revised version of the manuscript.

It may also be worth mentioning the evolutionarily conserved organization of the TCRd locus within the TCRA locus from zebrafish to mice and if possible better compare rearrangement properties of the TCRd locus (which already evolved D segments) to that of its host TCRA locus in zebrafish.

Response: We thank the reviewer for this suggestion. In a previous communication (Giorgetti et al., Sci. Advances 7, eabd8180, 2021), we have described *Tcra* and *Tcrd* repertoires of a teleost, *P. progenetica*, that was also included in the teleost group of species here. In **new Supplementary Figure 6**, we provide an analysis of the sequence signatures of *Vd*, *Dd* and *Jd* elements. The *V* elements dominating in *tcrd* assemblies (6/54 in the *Va/Vd* cluster) are almost never found in *tcra* assemblies. Moreover, they exhibit no recognizable shared sequence pattern next to the RSS, clearly distinguishing them from a typical *Va* element. We note that the two *Jd* elements also do not share sequence similarity next to the RSS, nor to the ends of the *Vd* or *Dd* segments. These results suggest that, although *Vd* elements are scattered across the *Va/Vd* cluster, they lack the 'primordial' signatures, in line with the fact that *tcrd* assemblies not only incorporate D elements, but also exhibit a significant number of non-templated nucleotides as described in our previous study (Giorgetti et al., Sci. Advances 7, eabd8180, 2021).

10. Some words and phrases are vague throughout the manuscript. Some specific examples: In the discussion, the sentence in lines 281-282 is not correct. Thus, TdT expression in fact increases overall recombination diversity and does not “diminish overall recombination diversity”.

Response: We thank the reviewer for spotting this howler. Of course, it is the opposite of what was written. We have corrected the text accordingly.

Also, as mentioned, TdT expression suppresses “recurrent or dominant” recombination that involves microhomologies in the RSS-proximal regions of coding ends being joined, but does not necessarily block NHEJ using microhomologies per se.

Response: Thank you, we have toned down and rephrased the relevant statement in the discussion section.

Other examples are in line 269, in which it is not clear what exactly 'epistasis' refers to as used, and in line 288, the phrase 'In this situation is unclear (which situation?)

Response: In the original version, 'epistasis' referred to identical/similar TSD-derived microhomologies. To clarify our meaning, we have replaced 'epistasis' by 'TSD-derived microhomologies'.

'In this situation....' referred to the fact that some *Vg-Jg* rearrangements are seen in both fetal and adult gd T cells. What we meant to say is that when a particular V and J combination is used at two stages that differ with respect to TdT expression, the microhomologies must be maintained, otherwise stage-specific stereotyped rearrangement becomes impossible. We have rephrased this section to make our intention clear.

Referee #2:

The authors propose that the ancestral RAG recombination process occurs via microhomologous recombination. This would have avoided the risk of harmful self-recognition that would have occurred without a mechanism of negative selection (via the thymus for example) that was probably lacking in the vertebrate ancestor. The region of microhomology corresponds to the TSD generated by the transposon RAG, which has been domesticated in jawed vertebrates as the RAG gene.

The analysis presented by the authors on the TCR alpha repertoire across jawed vertebrates is consistent with the hypothesis. They show that recombination of the RAG process occurs via microhomologous recombination in cartilaginous vertebrates, bony fishes but not in tetrapods. As cartilaginous vertebrates and paraphyletic bony fishes (along with tetrapods) conclude that the ancestral state of recombination occurs via microhomologous recombination.

Comments to authors

1) Could the authors better explain why they think recombination via microhomologous regions is ancestral? I'm not sure non-specialists would understand.

Response: We thank the reviewer for alerting us to this issue. Accordingly, we have attempted, within the limited space available, to better articulate our basic hypothesis that the TSD is the basis for microhomology-directed recombination and that with the emergence of TdT (a vertebrate-specific paralog of DNA polymerase mu) new

opportunities arose for the diversification of V-J (and also D-J and V-DJ) junctions. This has been mentioned in the revised discussion section and elaborated upon in the **new Supplementary text**.

2) The authors have written in several places that microhomologous recombination is due in part to the inefficiency of TDT (Terminal Deoxyribonucleotidyl Transferase). Microhomologous recombination occurs in the case of the zebrafish T-cell receptor alpha. As far as I understand, this is not the case for the zebrafish T-cell receptor beta (please clarify this point). If so, is it possible that TDT is not expressed between the DN4 stage and the double-positive stage (where the V α J α recombination occurs) and is expressed in the stage where the V β J γ recombination occurs?

Response: The reviewer is correct. *TCRB* assemblies often show signs of non-templated nucleotide additions, as we have stated in the manuscript. Unfortunately, it is currently not possible to isolate teleost thymocytes that would mirror the successive stages of mouse T cell development, if they at all exist in a 1:1 equivalence. Hence, we have no way of determining TdT activity in what would be the equivalent of the DN4 stage; note also that there is - to the best of our knowledge - no evidence yet of a pre-Ta gene in teleosts, indicating a different developmental trajectory for fish T cell development. Moreover, it is not clear, when *Tcrg/d* rearrangements take place during thymocyte development of zebrafish. But it is clear that the *Vb-Db-Jb* (this is probably what was meant) junctions are modulated by TdT activity.

If TDT is not expressed between the DN4 and double positive stages (where V α J α recombination occurs) how do the authors explain the "rogue" elements Va16-1 and Ja5?

Response: The reviewer raises an interesting point. We reported that the usage of the "rogue" elements is very low; we attribute this to the requirement for more extensive "nibbling" at the free coding ends to arrive at an in-frame fusion than would be required for "ordinary" elements.

3) Page 141: Several transposons closer to RAG than to Transib have been described. The TSD is the same size as in the case of Transib (PMID: 27293192).

Response: We thank the reviewer for this comment. The appropriate reference has been inserted.

4) Even if the TDT is absent after the excision of the intermediate DNA, hairpins are formed in the surrounding DNA. Next, the hairpins are treated with Artemis endonuclease, effecting an asymmetric opening of the hairpin and leading to a palindromic variation of P nucleotides at the junction. This is described (of course) not only in the case of RAG but also in the case of the transposon RAG (PMID: 27293192). Do the authors see this (P nucleotide diversity) in their analysis on the TCR alpha repertoire (even if this is found in few cases)?

Response: We thank the reviewer for raising this point. Indeed, in Fig. 2b, we indicate the presence of microhomology-based recombination based on the generation of a P nucleotide (pattern 4); this fact is further elaborated with data shown in **new Extended Data Figure 10e,f**.

Minor comments

5) L255: What do the authors mean by "The radical deviation with Darwinian selection" (it seems to me that they mean substitutions versus recombination)? Please clarify.

Response: We thank the reviewer for alerting us to this possible misunderstanding. We have modified the text as follows:

' Germline-encoded antigen receptors, which underlie self/nonself discrimination in innate immune system, are subject to Darwinian selection over evolutionary time. The sudden emergence of the facility of somatic recombination in ancestral vertebrates raises the question of how they struck the balance between immunologically desirable sequence diversity of receptors and suppression of potentially catastrophic autoimmunity through inadvertent recognition of self.'

6) L274: In the case of iNKT and MAIT cell receptors, does V(D)J recombination occur via the microhomologous regions?

Response: This is an excellent point and thank you for alerting us to this issue. For the cognate *Va-Ja* rearrangements that are found in these "innate-type" of T cells, it appears that micro-homology is not involved; instead, they instruct lineage-specific differentiation after agonist selection from a randomly generated repertoire. What he had intended to say was that **stereotyped rearrangements** (irrespective of their mechanism of generation) can be functionally advantageous, but we now realize that this may be confusing in the way we presented this argument. Space constraints prevent us from elaborating this line of reasoning to the extent that it makes the distinction clear; we have therefore decided to omit this point from the revised manuscript.

7) Recombination via the regions of microhomologies, after transposon excision creating hairpins, has been described (see for example PMID: PMC321453). It could be the same for the RAG transposon. This should be discussed in the context of the authors' results.

Response: We thank the reviewer for alerting us to this study. It is indeed an interesting example for what we propose to have happened at the onset of RAG-mediated recombination. We have added this reference in the discussion section of the revised manuscript.

Referee #3:

The article submitted by Giorgetti, O'Meara, Schorpp, and Boehm provides a new understanding of the evolution of adaptive immunity. They make the novel observation (as far as I know), that most TCR-alpha VJ rearrangements in zebrafish arise via microhomology-directed, non-homologous end joining, and thus, even in the absence of selection, alpha-chain VJ junctions are largely in-frame. In addition, there are almost no non-templated sequences in the mature repertoire of zebrafish. Both of these observations are consistent with a lack of TdT expressed during the process of zebrafish alpha-chain rearrangements. This means that the TCR a-chain in zebrafish is limited in diversity, in

stark contrast to the parallel analysis carried out on mice. Since there is a TdT ortholog expressed in the thymic cortex of zebrafish, could the authors comment on the lack of an apparent TdT influence?

Response: Thank you for raising the issue of TdT activity in zebrafish and other teleosts. We agree that the most likely explanation for the difference between *tcrα* and *tcrβ* assemblies is the presence of absence of TdT (or perhaps DNA pol μ) during the assembly process. Unfortunately, the developmental progression of T cells (and B cells for that matter) in zebrafish is not well defined and it is thus currently impossible to isolate teleost thymocytes that would mirror the successive stages of mouse T (or B) cell development. Hence, although important, the analysis of this question is not trivial, and must await the development of stage-specific isolation procedures and is thus beyond the scope of our manuscript. As mentioned by the reviewer, others have demonstrated differential expression of TdT by in situ hybridization of thymus sections (Beetz et al., *Immunogenetics* 59, 735, 2007); more recently by scRNA-seq data for T cells (and B cells) have been reported (Rubin et al., *J. Exp. Med.* 219, e20220038, 2022). In sum, these observations suggest that the developmental regulation of TdT may be a conserved feature of lymphocyte development. We have introduced reference to these two studies in the revised version of the manuscript.

The authors show this by analyzing zebrafish that have a heterozygous mutation in the alpha locus such that one allele is functional and the other undergoes recombination with no possible cellular selection. They refer to this as the non-selectable repertoire, and show that it is almost entirely in-frame. When they displaced the position of the RSS elements, this in-frame pattern disappeared showing that the position of a microhomology is essential to the conservation pattern.

The analysis goes to show that these observations are true for three other teleost species, and in a wider analysis, the conserved microhomologies are present in cartilaginous, basal, and teleost fishes, whereas it is missing lungfish and mammals. The authors propose that a "TG" motif is part of a primordial sequence pattern that directs alpha-chain gene assembly in cartilaginous and basal fishes, and teleosts.

All of these very extensive analyses are very well documented in extensive supplementary material and clearly presented in clean (and sometimes colorful) expository writing.

Based on these findings, they present a theoretical answer to a conundrum of the adaptive/acquired immune system that is, how would a highly diverse repertoire of lymphocyte receptors coincidentally evolve with mechanisms to ensure self-tolerance? Their novel hypothesis is that the adaptive immune system in the proto-vertebrate (both gnathostomes and agnatha) was restricted in sequence diversity, and presumably most of the resulting receptors were not specific for self-molecules present in the species. With time, cellular tolerance mechanisms provided a selective advantage, even to the protovertebrates, that eventually allowed sequence diversity to evolve to that is found in birds, skates and mammals.

A prediction, based on the authors' theory of limited diversity and self-tolerance, is that in the protovertebrate, both alpha and beta sequences would be restricted since of course the beta chains can contribute equally to antigen peptide specificity. For some reason this limitation in alpha chain diversity has been maintained in zebrafish for the alpha chain, but it appears to have evolved in the b-chain locus to look like the diversity found in mammals (Fig. 1e; Supplementary Tables 1,3). At least that was my initial take; however, in figure 3E, the ordinate is "mean microhomologies per molecule", whereas the text describes non-templated nucleotide insertions". Could the authors please clarify this for me and indicate whether the zebrafish b-chain locus shows evidence of restricted diversity and a lack of TdT (and again the presence of thymic TdT)?

Response: We thank the reviewer for raising this point. We agree that we should have been more consistent in our description. We infer a dominant role for microhomologies in situations where there are few, if any, non-templated nucleotides and a common cross-over point between *V* and *J* sequences. In retrospect, using the term 'mean microhomologies per molecule' was perhaps confusing. What was meant was how many identical nucleotides from the germ-line encoded *V* and *J* elements are found in the respective *V-J* joints. We have changed 'mean microhomologies per molecule' to 'mean length of microhomology per molecule' to provide a more precise description of our metric. The purpose of Fig. 3e was to demonstrate the difference in junctional diversity not only among the *tcra* assemblies of different species, but also to contrast it to the much more diverse *tcrb* repertoire. We attribute these differences to developmentally regulated and species-specific activities of TdT.

Why have the teleosts retained the protovertebrate VJ recombination and repair mechanism? For example, do zebrafish have stunted mechanisms of tolerance (if this is known)?

Response: Thank you for raising this point. We cannot offer a definitive answer to this question, but what we can say is that teleosts possess genes encoding key elements of recessive (central) and dominant (peripheral) tolerance, Aire and Foxp3 respectively. But it may well be that additional tolerance mechanisms that we know from mammals may not be present in teleosts. Further work focusing on the functional role of genes like Aire and Foxp3 in immune tolerance in teleosts and cartilaginous fishes is required to clarify this issue.

Another issue is B cell receptors and T cell receptors. Do the antibody loci in teleosts also show restricted diversity, or have these antibody-encoding loci evolved to show high levels of diversity and alongside effective tolerance mechanisms?

Response: We know from our own studies (Giorgetti et al., Sci. Advances 7, eabd8180, 2021) and the work of others (Weinstein et al., Science 324, 807, 2009) that *IgH* genes of teleosts exhibit a high degree of CDR3 diversity. Less is known about *IgL* genes in teleosts. In a publication from 2008, Zimmerman et al. (described the genomic configuration of *IgL* chain genes in zebrafish (Dev. Comp. Immunol.32, 421, 2008). In the revised manuscript, we provide alignments of *V_L* and *J_L* elements for the five identified genomic clusters (**new Supplementary Fig. 4**). This analysis shows the presence of similar sequence motifs at the ends of some *V* and *J* elements; the sequences

of several V-J assemblies provided by Zimmerman et al. suggest that these may impact the outcome of assemblies, although more work is required to clarify this issue.

In the Discussion, there is a statement, “Although TdT has been shown to suppress microhomology-guided recombination and diminish overall recombination diversity^{29,32-34}, which in fetal life leads to a more stereotypical repertoire...” What is meant by diminish the overall recombination diversity? I would think that TdT increases the overall receptor diversity? Can the authors please clarify.

Response: We thank the reviewer for spotting this howler, which has also been recognized by reviewer 1. Of course, it is the opposite of what was written. We have corrected the text accordingly.

Overall, this is a spectacular set of findings and accompanying analyses. I think the theory is novel and interesting, and perhaps testable. I highly recommend publication in Nature.

Response: We thank the reviewer for her/his positive evaluation of our work.

Referee #4:

The manuscript by Giorgetti et al. is an intriguing exploration of the hypothesis that tiny stretches of DNA sequence homology near the ends of recombining segments used in V(D)J recombination were key in allowing the emergence of such a stochastic system of receptor generation without overly detrimental self-recognition. Here they present a robust set of comparative genomic analyses of the T cell receptor alpha loci of diverse species that supports the hypothesis. Additionally, they find that TCR alpha repertoire diversity as expressed at the mRNA level correlates with the extent of this microhomology.

The authors test this hypothesis with a great deal of work, including CRISPR-engineered zebrafish, a remarkably wide comparative approach involving sampling everything from sharks to elephants, repertoire sequence analysis in species that must have required significant troubleshooting of amplification parameters, and a Herculean amount of genomic and amplicon bioinformatics. Even with the IMGT database and the most cutting-edge tools, those of us who look at antigen receptor loci for a living know that assemblies in these regions are wrought with pitfalls... there is no way around much manual analysis, checking and annotation.

Twenty years after Tonegawa demonstrated somatic gene rearrangement in lymphocyte antigen receptors, Sam Schluter and Jack Marchalonis described the “Big Bang” that appeared to be the genesis of our combinatorial adaptive immune system (Schluter et al., *Dev. Comp. Immuno.*, 1999) in sharks. Nearly twenty years later David Schatz and team definitively showed us how the horizontal transposon event evolved into the RAG recombinase structurally (Liu et al., *Nature*, 2019). Now, for the first time, that Big Bang seems clear... and a bit more of a rolling start?

The steps are now much more clear how on top of the historically-gazing, reactive, innate immune system a radically preemptive, forward-gazing, anticipatory system evolved.

This is likely not an easy read, even for those somewhat versed in the immunogenetics of lymphocyte antigen receptors and theories on their origins. However, ample background is not possible in the format. The following items I think bear consideration before this manuscript is ready for the readership of Nature.

Response: We thank the reviewer for her/his thoughtful and encouraging evaluation of our work. We do agree that we address a complex question, and have attempted to address the points raised below by rephrasing several parts of the manuscript, all within the tight space restrictions imposed by the article format, and by adding a **supplementary text** to elaborate on some of the concerns/questions that might be shared also by other readers.

Major points

The more impactful the work the longer the list of obvious new questions and experiments mandated, but I know a line has to be drawn somewhere. These first two major questions many readers will have spring from the focus on the alpha chain, with little explanation why. While I do not think deeper exploration of other antigen receptor loci is necessary for this paper, I do hope a little discussion and potential next hypotheses can be incorporated into the tight word limits. It will be important to help the general reader see the broader impact of this work.

1) In the first paragraph of the introduction the authors acknowledge that the hypothetical gene targeted by the ancestral (transib) transposon is considered the founding member of the AgR of jawed vertebrates, yet alpha is studied instead of gamma (or even immunoglobulin light chain lambda), which is sometimes suggested to be the ancestral chain of our lymphocyte antigen receptors. Were loci encoding other chains than TCR alpha and beta explored? The work on TCR alpha is deep and complete (my second paragraph above), which is why I wonder why the jump to this locus instead of TCR gamma which would be simpler in many ways (fewer V and J elements, no confounding delta use (line 232)). It is recognized that the D segment using loci (immunoglobulin heavy chain, TCR beta and delta) are not the most parsimonious candidates for the ancestral.

Response: The reviewer raises an important point, which we have attempted to resolve by composing a supplementary text to explain in more detail why we have selected the *tcr* chain gene as a reasonable starting point to explore the TSD/diversity hypothesis. Among extant vertebrates, three groups of genes retain the presumed primordial V-J structure: *TCRa*, *TCRg*, and *IgL* gene loci. Although it is impossible to decide which of the V-J antigen receptors can rightfully claim ancestral priority, we focused on *TCRa* for several reasons:

(1) The *TCRg* locus often consists of only very few V and J elements. For instance, in the mouse, 7 *Vg* and 4 *Jg* elements are recognized; in the zebrafish, 7 *Vg* and 7 *Jg* are known. By contrast, the *TCRa* locus usually comprises dozens of V and J elements (mouse: 64 *Va* and 45 *Ja* elements respectively; zebrafish: 124 *Va* and 134 *Ja* elements respectively).

Based on the data deposited in the IMGT database (<https://www.imgt.org/>), differences in the numbers of *V* and *J* segments between *TCRg* and *TCRa* also hold for 11 other species, as depicted in **new Supplementary Figure 1a**.

(2) It has now been shown for many species (such as squamates [Morrissey et al., *J. Immunol.* 208, 1960, 2022]) that they have lost the genes for the *TCRgd* receptor. Apart from introducing bias in broad phylogenetic surveys, the statistical robustness of any conclusion about conserved sequence elements is inversely proportional to the numbers of *TCRg*-associated *V* and *J* elements. Hence, we consider the inferences derived from the analyses of the *TCRa* loci to be less prone to random fluctuations and therefore more meaningful.

With respect to the analysis of *IgL* genes, two reasons suggested to us that they may not be the best objects of study at this point in time:

(1) *IgL* loci also commonly present with few *J* elements, although the number of *V* elements varies considerably (**new Supplementary Fig. 1b**). By way of example, a total of 51 *V_L* genes and 17 *J_L* (all 5 loci combined) were described for zebrafish (Zimmerman et al., *Dev. Comp. Immunol.* 32, 421, 2008), and 146 *V_L* and 12 *J_L* for the mouse (kappa and lambda loci combined; IMGT database).

(2) Since the number and structure of *IgL* gene loci varies among vertebrate species, the resulting ambiguity complicates the analysis when it comes to definition of orthology. In sum, we considered *TCRa* to be the best target for our investigation, as it is a constant companion of canonical adaptive immune systems in jawed vertebrates (with the exception of a small number of species of deep-sea anglerfishes that we recently found to have lost canonical adaptive immunity after pseudogenization of RAG genes [Swann et al., *Science* 369, 1608, 2022]).

2) Related to point #1 above... This evolution of the TCR alpha V and J arrays presumably was in lock step with the MHC allelic “repertoires” in these early vertebrates. To my knowledge, restriction of canonical alpha/beta T cell receptor to MHC peptide presentation is extremely early in the system... MHC is both polygenic and polymorphic in all extant cartilaginous jawed fishes, and even lamprey VLRs may have a receptor and lymphocyte subset transcriptionally orthologous to gnathostome alpha beta T cells. Do the authors see the CDRs 1 and 2 of the TCR alpha variable segments quickly evolving for recognition of the alpha helices of the MHC peptide binding domains and the junctional diversity of CDR3 for the peptide?

Response: The reviewer raises an interesting question. If we understand correctly, an investigation into this problem would require an extensive correlative study of MHC sequences and *TCRa* and *TCRb* sequences. This may be possible for species for which structural information on the oligomeric TCR/pMHC complexes is available, but would certainly be difficult to extend to species where such information is lacking. We therefore feel that this issue, though clearly interesting, is beyond the scope of the current study.

3) Perhaps too late now, but the title does not quite capture the gestalt of the paper. Isn't it suggesting the origin of tolerance in our adaptive immune system... and actually challenging the last of Sir McFarlane Burnet's four postulates of clonal selection? Perhaps autoreactive lymphocytes do not have to be negatively selected if they do not arise due to restricted junctional diversity.

Response: MacFarlane Burnet's forth postulate posits that lymphocytes expressing self-reactive AgR are eliminated during maturation. We don't think that our results violate this tenet, since MacFarlane Burnet developed his ideas from observations in mammals, and there they clearly hold. We share the reviewer's view on the implication of our current results. The key difference between a primordial state and the extant condition is the level of diversity: if it is high, clonal deletion (and/or dominant tolerance) is mandatory; if it is low, the immune system may get away without special tolerance mechanisms, as the receptor components are effectively selected by Darwinian forces acting upon microhomologies.

4) Is it possible that such a mechanism was not necessary? Seems the antigen universe of non-self/potential pathogen was likely larger than the self Ag repertoire to begin with, tolerance mechanisms would just be value added. Lines 47-50 may not be right, although the data here suggest it was advantageous for the development of our adaptive system half a billion years ago.

Response: To add to our thoughts concerning point#3 raised by the reviewer: Yes, we think of the primordial type of AgRs as innate-like, perhaps tuned to more general properties of potential non-self/pathogen-derived antigens. It may well be that the proliferative property of 'adaptive' lymphocytes was the initial advantage over a more stereotyped innate response. When it eventually became necessary to introduce fine-grained discrimination of antigenic structures, the risk of self-reactivity increased, with all the known consequences for efficient quality control. Space constraints unfortunately prevent us from elaborating on this important idea.

Minor points

- Are the four patterns of microhomology passed through speciation events and higher taxa? (Do the same V-J pairs with the same microhomologies exist between two teleosts, or between a shark and a lungfish?) I am questioning how species-specific the public repertoires actually are (line 173).

Response: We note that it is difficult to establish unambiguous orthology for individual *V_α* and more so for *J_α* elements, even within the teleost clade. We believe it is rather more interesting that even across teleost species separated by hundreds of millions years of independent evolution the sequence patterns have been preserved (the lineages giving rise to zebrafish and medaka separated over 200 million years ago [Hughes et al., PNAS 115, 6249, 2018]), clearly attesting to their functional importance for teleost immune systems. In **new panels to Extended Data Figure 10 (10e,f)** we now present data indicating that a small fraction of mouse *J_α* elements still exhibit the 'primordial' *J_α* pattern. Note also that our sequence analysis encompasses the entire VJ sequence, not just the CDR3 sequence, to provide unambiguous assignments for the usage of individual elements.

- AID-mediated somatic hypermutation has been described at nurse shark TCR alpha, occurring in the thymus (Ott et al., ELife, 2018). Does this present a problem for the junctional repertoire restriction mechanism described here? The TCR SHM could have evolved much later, at least one way to reconcile.

Response: This is an interesting point, which we could not discuss in the original version of the manuscript owing to space constraints. We agree with the reviewer's opinion that SHM was probably introduced as a subsequent diversification mechanism of *TCRa* chains. The reason we think this may be true is that the degree of conservation of *Va* and *Ja* in cartilaginous fishes is somewhat lower than in the presumed ancestral configuration. Given the space constraints, we haven't found a way to adequately address this point (and many other interesting implications of our study) and must leave it to the reader to consider them for the time being, and relegate an in-depth analysis to a separate communication.

- I am intrigued by the range of the amphibian RSS J frame conservation (line 241) and how it does not quite fit in with the trend (Fig. 4b). Any ideas as to why? Louis du Pasquier and others have studied the *Xenopus* immune system pre- and post-metamorphosis, a radical reboot. Perhaps it is related to that? And does it track with size (limited T cell compartment size in very small tadpoles of some species may limit?). I am curious if there is a relation across the range of species between body size and RSS conservation.

Response: The reviewer raises an interesting point. In fact, one of our initial motivations for the present study was the hypothesis that the extent of CDR3 diversity may scale with body size (that is, lymphocyte numbers), hence our inclusion of species with very different body sizes. For example, the body sizes of a mouse and an elephant differ by about 5 orders of, yet the analysis of *TCRa* junctions indicates that their entropy values - as a measure of potential diversity - do not show such difference. However, since our initial body size/diversity hypothesis was thus refuted by the data, the present differences must rather be a reflection of phylogenetic differences among vertebrate clades. As pointed out by the reviewer, the frog (as a representative of the amphibian species in our collection) has been the focus of much work in the past, and the "elephant-tadpole paradox" is well known and much debated in the immunological literature. Whereas the changes to lymphocyte number and immune system reorganization during metamorphosis clearly merit a revisit, for instance using the type of analysis presented in our manuscript, we believe that it is beyond the scope of the present communication.

- Line 235, what is "c.f."?

Response: Thank you. This has been clarified.

- Line 289, the first "are" needs to be removed.

Response: Thank you. This has been done.

- Line 423, looks like an aberrant bold in "length".

Response: This typo is not present in the original word file.

- Line 570, last word should be "the" not "then".

Response: Thank you. This has been corrected

- Why were ten species excluded from Ext. Data Fig. 12?

Response: 10 species listed in Supplementary Table 4 (and analysed in all aspects) are excluded from the cladogram, because they are either designated as extinct in the Open Tree of Life database (such as coelacanth - N.B.: we notified the curators to correct this oversight), or have alternative species designations.

- Fig. 3a, would be useful to align the amino acid motif under or above the nucleotide seqs for orientation (C of the V, how far 5'/amino-terminal of the FGxG of the J).

Response: Thank you. We have chosen to give relevant information in the legend, since the degenerate nucleotide sequences and the variable distances to the conserved phenylalanine residue would have made the labelling rather cumbersome.

Reviewer Reports on the First Revision:

Referees' comments:

Referee #1:

The authors have fully addressed all of my comments with revisions, modifications and a supplementary discussion that fully clarifies a number of points I raised. The manuscript now reads very well. The abstract now perfectly describes the content and points previously unclear in the text have been clarified. I only hope that interested readers will be clearly led to the Supplementary Discussion. This is an important and very interesting study that should be of broad interest. I congratulate the authors for, as I can tell, attending to the comments of all four reviewers and producing a terrific revised manuscript.

I strongly recommend publication.

Fred Alt

Referee #2:

Q 1) Could the authors better explain why they think recombination via microhomologous regions is ancestral? I'm not sure non-specialists would understand.

Response from the authors: We thank the reviewer for alerting us to this issue. Accordingly, we have attempted, within the limited space available, to better articulate our basic hypothesis that the TSD is the basis for microhomology-directed recombination and that with the emergence of TdT (a vertebrate-specific paralog of DNA polymerase mu) new opportunities arose for the diversification of V-J (and also D-J and V-DJ) junctions. This has been mentioned in the revised discussion section and elaborated upon in the new Supplementary text.

The analysis presented by the authors on the TCR alpha repertoire across jawed vertebrates is consistent with the hypothesis. They show that recombination (which follows DNA excision via RAG) occurs via microhomologous recombination in cartilaginous vertebrates, bony fishes but not in tetrapods. As cartilaginous vertebrates and bony fishes are paraphyletic (tetrapods and bony fishes are sister species while cartilaginous vertebrates form an outgroup), one can conclude that in the ancestor of jawed vertebrate, the RAG recombination occurred via microhomologous recombination.

I think that this is a key argument.

Referee #3:

My queries were mainly informational, and the authors answered them appropriately. Having read through all the reviewers comments and the authors responses, I am even more firmly of the opinion that this is a paper suitable for publication in Nature.

Referee #4:

I am good with the revision. It is a significant improvement and worthy of Nature. But some thoughts:

A lot of the old and new reasons for not exploring potential ancestral AgR IgL or gamma still resonate as "looking for your keys under the streetlight"... it is much easier to search there. Still it is good that this rigorous analysis of alpha has been tackled. Mark Davis and others have suggested that it might be the ancestral receptor, and just getting through positive selection is a monumental task (requiring many J's), pre-check beta with the surrogate alpha, and as Reinherz recently showed checking with MHC as early as DN3. Hunkapiller (1993) noticed the J's in non-rearranging V domain encoding genes, RAG allows the thrust for combinatorial diversity AND CDR3 length.

Whether this is truly ancestral, an evolutionary driver, or accessory is debatable, but still pretty cool.

Author Rebuttals to First Revision:

Response to Reviewers' comments

Referee #1:

The authors have fully addressed all of my comments with revisions, modifications and a supplementary discussion that fully clarifies a number of points I raised. The manuscript now reads very well. The abstract now perfectly describes the content and points previously unclear in the text have been clarified. I only hope that interested readers will be clearly led to the Supplementary Discussion. This is an important and very interesting study that should be of broad interest. I congratulate the authors for, as I can tell, attending to the comments of all four reviewers and producing a terrific revised manuscript.

I strongly recommend publication.

Fred Alt

Response:

We are grateful to Dr Alt for his constructive critique of the first version of the manuscript that helped us to considerably strengthen the revised version, and thank him for his encouraging comments on the second version.

Referee #2:

Q 1) Could the authors better explain why they think recombination via microhomologous regions is ancestral? I'm not sure non-specialists would understand.

Response from the authors: We thank the reviewer for alerting us to this issue. Accordingly, we have attempted, within the limited space available, to better articulate our basic hypothesis that the TSD is the basis for microhomology-directed recombination and that with the emergence of TdT (a vertebrate-specific paralog of DNA polymerase mu) new opportunities arose for the diversification of V-J (and also D-J and V-DJ) junctions. This has been mentioned in the revised discussion section and elaborated upon in the new Supplementary text.

The analysis presented by the authors on the TCR alpha repertoire across jawed vertebrates is consistent with the hypothesis. They show that recombination (which follows DNA excision via RAG) occurs via microhomologous recombination in cartilaginous vertebrates, bony fishes but not in tetrapods. As cartilaginous vertebrates and bony fishes are paraphyletic (tetrapods and bony fishes are sister species while cartilaginous vertebrates form an outgroup), one can conclude that in the ancestor of jawed vertebrate, the RAG recombination occurred via microhomologous recombination.

I think that this is a key argument.

Response:

We thank the referee for re-emphasizing this point. We have introduced a statement to the effect at the end of the results section to make this argument more explicit.

Referee #3:

My queries were mainly informational, and the authors answered them appropriately. Having read through all the reviewers comments and the authors responses, I am even more firmly of the opinion that this is a paper suitable for publication in Nature.

Response:

We thank the referee for the encouraging and positive assessment of our work.

Referee #4:

I am good with the revision. It is a significant improvement and worthy of Nature. But some thoughts:

A lot of the old and new reasons for not exploring potential ancestral AgR IgL or gamma still resonate as "looking for your keys under the streetlight"... it is much easier to search there. Still it is good that this rigorous analysis of alpha has been tackled. Mark Davis and others have suggested that it might be the ancestral receptor, and just getting through positive selection is a monumental task (requiring many J's), pre-check beta with the surrogate alpha, and as Reinherz recently showed checking with MHC as early as DN3. Hunkapiller (1993) noticed the J's in non-rearranging V domain encoding genes, RAG allows the thrust for combinatorial diversity AND CDR3 length.

Whether this is truly ancestral, an evolutionary driver, or accessory is debatable, but still pretty cool.

Response:

We thank the referee for re-emphasizing the evolutionary aspect of our work and are grateful to her/him for the positive evaluation.

In response to the comment, we amended the Supplementary text to alert the reader to the fact that some immune-related molecules possess sequences similar to J elements C-terminal to a V-like Ig domain. To the best of our knowledge, Williams and colleagues described the first striking example when they deduced the sequence of the CD8 beta chain; moreover, in their report, they noted high sequence similarity of the V-like region to Ig lambda V sequences (Nature 323, 74, 1986). These findings support the idea that rearranging and non-rearranging Ig domain containing immune-related proteins have a common evolutionary origin.